# LatentLens: Revealing Highly Interpretable Visual Tokens in LLMs

**Benno Krojer** [1 2]  **Shravan Nayak** [1 3]  **Oscar Mañas** [1 3]  **Vaibhav Adlakha** [1 2]
**Desmond Elliott** [* 4]  **Siva Reddy** [* 1 2 5]  **Marius Mosbach** [* 1 2]

## Abstract

Transforming a large language model (LLM) into a vision-language model (VLM) can be achieved by mapping the visual tokens from a vision encoder into the embedding space of an LLM. Intriguingly, this mapping can be as simple as a shallow MLP transformation. To understand why LLMs can so readily process visual tokens, we need interpretability methods that reveal what is encoded in the visual token representations at *every* layer of LLM processing. In this work, we introduce LatentLens, a novel approach for mapping latent representations to descriptions in natural language. LatentLens encodes a large text corpus and stores contextualized token representations for each token in that corpus. Visual token representations are then compared to these contextualized representations and the top-$k$ nearest neighbor representations serve as descriptions of the visual token. We evaluate this method on 15 different VLMs, showing that commonly used methods, such as LogitLens, substantially underestimate the interpretability of visual tokens. With LatentLens instead, the majority of visual tokens are interpretable across all studied models and all layers. More broadly, our findings contribute new evidence on the alignment between vision and language representations and open up new directions for analyzing the latent representations of LLMs.

🔍 LatentLens Demo     Code

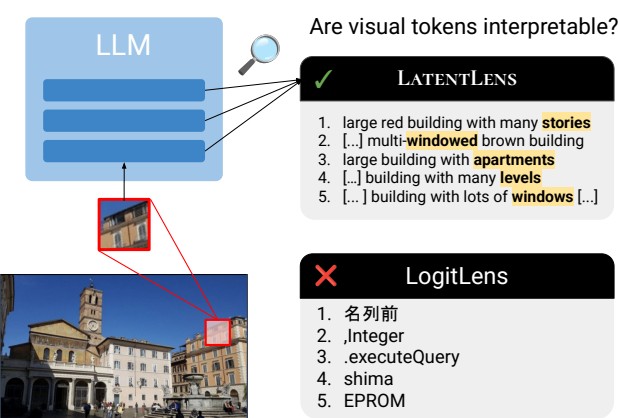

*Figure 1.* **Illustration of our method.** LatentLens compares latent representations of visual tokens to contextualized text representations obtained from full sentence descriptions.

*Equal senior contribution. [1]Mila – Quebec AI Institute, Montreal, Canada [2]McGill University, Montreal, Canada [3]Université de Montréal, Montreal, Canada [4]University of Copenhagen, Copenhagen, Denmark [5]Canada CIFAR AI Chair. Correspondence to: Benno Krojer <benno.krojer@mila.quebec>.

*Proceedings of the 43$^{rd}$ International Conference on Machine Learning*, Seoul, South Korea. PMLR 306, 2026. Copyright 2026 by the author(s).

## 1. Introduction

Transforming a large language model (LLM) into a vision-language model (VLM) can be as simple as training a linear transformation or shallow MLP that maps visual representations into the embedding space of a frozen LLM (Tsimpoukelli et al., 2021; Merullo et al., 2023; Liu et al., 2023; Beyer et al., 2024; Mañas et al., 2023, *inter alia*). While VLMs can be further improved via multiple stages of fine-tuning (Bai et al., 2023; Cho et al., 2026), the empirical success of frozen LLMs processing non-language inputs raises an important question: Why is it so easy to adapt an LLM to process data from other modalities? It has been hypothesized that LLMs are "universal computation engines" that can process arbitrary modalities with minimal adaptation (Lu et al., 2022; Shen et al., 2023; García-de Herreros et al., 2024). Through training on vast amounts of natural language, LLMs may implicitly learn about the physical world, form priors about visual properties (Han et al., 2026), or even learn a more sophisticated world model (Patel & Pavlick, 2022). It has also been argued that separately trained vision and language representations may converge to a shared structure (Huh et al., 2024), which could facilitate training simple projections between them.

However, these hypotheses do not explain how visual rep-

resentations are integrated inside an LLM and its representation space. Are the visual tokens processed by an LLM interpretable, i.e., do their representations correspond to semantically meaningful language? Existing work suggests that visual tokens are rarely interpretable at the input level via nearest neighbors in the language model embedding space (Mokady et al., 2021; Neo et al., 2025, EmbeddingLens). Recent work has explored the utility of LogitLens, which uses the LLM unembedding matrix (nostalgebraist, 2020), to interpret and analyze visual token representations (Neo et al., 2025; Jiang et al., 2025b). Finally, training-based methods such as sparse autoencoders (Huben et al., 2024; Venhoff et al., 2025) or supervised probing (Fu et al., 2025) have also been applied to understand what visual representations encode. However, both embedding-space and training-based methods are inconclusive about the interpretability of visual tokens in LLMs.

In this work, we propose LatentLens, a novel interpretability method for analyzing latent representations in VLMs. LatentLens is training-free and provides fine-grained sentence-level descriptions of latent representations. **Our key insight is that the most natural comparison for visual token representations are *contextual* text representations**, and not the LLM embedding or unembedding matrix. Thus, LatentLens compares visual token representations to their nearest neighbors in a large pool of contextualized token representations from intermediate LLM layers. For example, in Figure 1, the visual token of a building yields the nearest neighbor **stories** in the context of "... building with many **stories** ".

Empirically, we analyze the interpretability of visual token representations across layers for 15 different VLMs. We find that compared to other training-free approaches (LogitLens and EmbeddingLens), LatentLens reveals consistently high interpretability, highlighting that previous methods substantially underestimate the interpretability of visual token representations. Averaged across all 15 VLMs and all layers we study, LatentLens renders 68% of visual tokens interpretable (according to a VLM-judge). When using EmbeddingLens and LogitLens, however, only 32% and 24% of visual token representations are labeled as interpretable, respectively. Through ablation studies, we provide additional insights about the nature of this interpretable alignment, showing, e.g., that even linear projections lead to interpretable visual token representations. We also find evidence for a *Mid-Layer Leap* in the learned projection: the visual token representations at the input and early layers align most strongly to contextualized representations from mid-layers (e.g., layers 8–16), suggesting that the learned projection targets semantic rather than lexical representations. We also present qualitative analyses showing that LatentLens produces rich sentence-level descriptions, unlike LogitLens which might return subwords or next-token predictions.

Overall, our findings challenge existing assumptions about the interpretability of visual tokens, and offer new insights about the alignment of vision and language representations. We encourage the reader to explore the interactive demo as well as our pip-package with easy access to our database of contextual embeddings to facilitate adoption.

## 2. Background

We first introduce technical background on VLMs and describe prior work on analyzing their latent representations.

### 2.1. Turning LLMs into VLMs

A common approach for converting an LLM into a VLM is to project representations produced by a vision encoder into the embedding space of the LLM via a learned connector. More formally, let `venc` be a pre-trained vision encoder, which produces a sequence of *image embeddings* $\text{venc}(x_{\text{img}}) = [\mathbf{v}_1, \mathbf{v}_2, \ldots, \mathbf{v}_{T_v}]$ with $\mathbf{v}_i \in \mathbb{R}^{d_v}$ for a given image $x_{\text{img}}$, let `llm` be a pre-trained language model, and let $x_{\text{text}}$ be a textual input. A multimodal input is constructed as follows: First, $x_{\text{text}}$ is processed by a tokenizer `tok` and then converted into token embeddings via an embedding matrix $\mathbf{E}_{\text{emb}} \in \mathbb{R}^{|\mathcal{V}| \times d}$. Second, *visual tokens* representing $x_{\text{img}}$ are obtained by projecting each image embedding $\mathbf{v}_i$ into the embedding space of `llm` using a projection function[1] $\text{proj} : \mathbb{R}^{d_v} \mapsto \mathbb{R}^d$. Finally, the visual and textual representations are concatenated $\mathbf{x} = [\mathbf{p}_1, \ldots, \mathbf{p}_{T_v}, \mathbf{e}_1, \ldots, \mathbf{e}_{T_t}]$, where $\mathbf{p}_i = \text{proj}(\mathbf{v}_i) \in \mathbb{R}^d \ \forall \ i = 1, \ldots, T_v$ and $\mathbf{e}_1, \ldots, \mathbf{e}_{T_t} = \text{emb}(\text{tok}(x_{\text{text}})) \in \mathbb{R}^{T_t \times d}$. The resulting multimodal sequence $\mathbf{x}$ is then processed by the `llm` and converted into latent representations $\mathbf{h}_i^{(\ell)} \in \mathbb{R}^d$, where $i$ denotes the position in the sequence and $\ell$ a layer, respectively.[2] During the forward pass, textual representations can self-attend to the information encoded in the visual representations $\mathbf{h}_i^{(\ell)}$.

**Training.** Given a dataset $\mathcal{D} = \{(x_{\text{img}}, x_{\text{text}})_i\}_{i=1}^N$ of image-caption pairs, the `llm` can be trained by minimizing the cross-entropy loss over the target caption tokens:

$$\mathcal{L}_{\text{LM}} = - \sum_{t=T_{\text{instr}}+1}^{T} \log p(y_t \mid y_{<t}, x_{\text{img}}),$$

where: $\text{tok}(x_{\text{text}}) = [\underbrace{y_1, \ldots, y_{T_{\text{instr}}}}_{\text{instruction}}, \underbrace{y_{T_{\text{instr}}+1}, \ldots, y_T}_{\text{caption}}]$, and, unless stated otherwise, only the weights of `proj` are trained, with the `llm` and `venc` weights frozen.

---

[1]In practice, `proj` is usually a linear layer, an MLP, or an attention-based module (Liu et al., 2023; Wang et al., 2024).

[2]Color-coding is used for visual token representations ($\mathbf{h}_i^{(\ell)}$).

## 2.2. Interpreting VLM representations

An important open question is what exactly is encoded in visual token representations as they are processed by an LLM, and to what extent are the latent visual token representations ($\mathbf{h}_i^{(\ell)}$) interpretable as language-like tokens? We will briefly introduce popular approaches for studying these questions. We note that other supervised or prompting-based approaches exist such as probing (Belinkov, 2022), SAEs (Huben et al., 2024), LatentQA (Pan et al., 2024) or Patchscopes (Ghandeharioun et al., 2024). Instead, we focus on methods which are training-free and directly leverage the LLMs representation space.

**EmbeddingLens.** The simplest approach to interpret visual token representations is to compare them to the elements of the embedding matrix $\mathbf{E}_{\text{emb}}$ of `llm`. This is motivated by the projection layer being trained to output representations that are compatible with the embedding space of `llm`. For this approach, each $\mathbf{p}_i$ (or its corresponding latent $\mathbf{h}_i^{(\ell)}$ at layer $\ell$) is compared to the embeddings in `llm`'s embedding matrix $\mathbf{E}_{\text{emb}}$ via cosine similarity and the top-k most similar embeddings serve as "labels." EmbeddingLens has been previously used for interpreting soft prompts (Lester et al., 2021), visual tokens (Mokady et al., 2021; Jiang et al., 2025a) or speech (Ògúnrèmí et al., 2025).

**LogitLens.** A slightly different but conceptually very similar approach is the LogitLens (nostalgebraist, 2020). Instead of comparing latent representations directly to embeddings, LogitLens projects the latent representation to the unembedding space of the model by multiplying $\mathbf{h}_i^{(\ell)}$ with the unembedding matrix $\mathbf{W}_{\text{unemb}} \in \mathbb{R}^{d \times |\mathcal{V}|}$, obtaining a distribution over vocabulary items. Then, the top-k vocabulary items, i.e., those with the largest logits, are retrieved as "labels". LogitLens been widely used in previous work analyzing LLMs (Geva et al., 2022; 2023; Fierro et al., 2025), and has recently been adopted for VLMs (Shukor & Cord, 2024; Neo et al., 2025; Jiang et al., 2025b).

# 3. LatentLens

In the following, we provide a unifying perspective on existing training-free methods for interpreting latent representations and introduce LatentLens, a novel method for mapping latent representations to natural language descriptions.

## 3.1. Unifying existing lenses

EmbeddingLens and LogitLens share the same goal and setup: To map a latent representation $\mathbf{h}_i^{(\ell)} \in \mathbb{R}^d$ at position $i$ and layer $\ell$, to a natural language description. Here, we provide a unified perspective on both methods. Formally, let $\mathcal{C}$ be a set of candidate descriptions, where each description $d_j \in \mathcal{C}$ is associated with a vector $\mathbf{r}_j \in \mathbb{R}^d$. $\mathbf{h}_i^{(\ell)}$ is

mapped to a description $d_j$ in three steps:

> ① **Compute scores:** For each $\mathbf{r}_j$, compute a score using a similarity function $f : \mathbb{R}^d \times \mathbb{R}^d \mapsto \mathbb{R}$:
> $$s_j = f(\mathbf{h}_i^\ell, \mathbf{r}_j), \quad j = 1, \ldots, |\mathcal{C}|.$$
> ② **Select top-$k$ vectors:** $I_{i,\ell} = \arg \text{top-}k(\{s_j\}_{j=1}^{|\mathcal{C}|})$.
> ③ **Return descriptions:** $\{d_j : j \in I_{i,\ell}\}$.

The set of possible descriptions for EmbeddingLens and LogitLens is given by the language model vocabulary, i.e., $\mathcal{C} = \mathcal{V}$. The similarity function is either cosine similarity with the rows of the embedding matrix $\mathbf{W}_{\text{emb}}$ (EmbeddingLens) or the inner product with the output embedding matrix $\mathbf{W}_{\text{unemb}}$ (LogitLens). Finally, in both cases, the vocabulary items are sorted based on their similarity score and the top-$k$ elements are selected as a description of $\mathbf{h}_i^{(\ell)}$.[3]

**Limitations of prior lenses.** Under our unified framework, two limitations of EmbeddingLens and LogitLens become apparent: 1) The description set is limited to a model vocabulary $\mathcal{V}$, containing only (sub-word) tokens. 2) Latent representations $\mathbf{h}_i^{(\ell)}$ from different layers are always compared to the same reference vectors, which are either the input or output embeddings. However, it is unclear whether the output or input embedding space is the most natural representation space to compare against. For example, LogitLens tends to works best for later layers, closer to the models output embedding space, and reliability can vary significantly across models (Geva et al., 2021; Belrose et al., 2023).

## 3.2. Method

We propose LatentLens, a novel interpretability method for mapping latent representations to descriptions in natural language. The key idea of LatentLens is that **the natural point of comparison for latent representations are other contextualized LLM representations** to serve as potential nearest neighbors, i.e., a token in the *context* of a sentence. Additionally, we believe that limiting the set of descriptions to individual sub-word tokens is unnecessarily restrictive and instead propose to use a large corpus of multiple-token sequences, e.g., sentences, which provides semantically richer descriptions for interpretation.

Concretely, LatentLens works as follows (see also Figure 2): Let $\mathcal{C}$ be a (large) corpus of sentences and $\mathcal{M}$ be an LLM with $L$ layers. We construct a set of reference vectors $\mathcal{R}$ by encoding every sequence $d_j \in \mathcal{C}$ with $\mathcal{M}$ and storing the contextualized token representations $\mathbf{r}_{j,t}^{(\ell)}$ at every position $t$ of the sequence and every layer $\ell$ of the model. To analyze

---

[3]We note that both methods can be seen as unsupervised version of *linear probing* (Belinkov, 2022), where instead of learning a linear transformation $\mathbf{W} \in \mathbb{R}^{d \times k}$ using supervised data, one relies on the pre-trained embeddding or unembedding matrix.

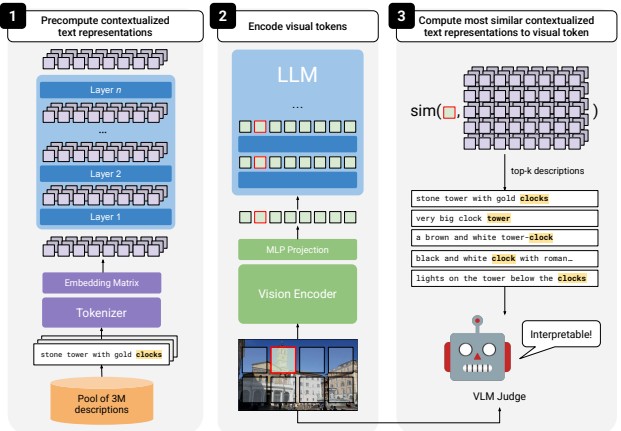

*Figure 2.* **Illustration of LatentLens.** (1) Contextualized token representations are precomputed in multiple layers of an LLM using a large corpus of descriptions. (2) Latent representations of visual tokens are extracted from all layers of the LLM, and (3) compared against the precomputed contextualized token representations. The interpretability of the visual token based on its top-$k$ descriptions can be automatically evaluated by a VLM-judge.

the latent representation $\mathbf{h}_i^{(\ell')}$ of a visual token at position $i$ and layer $\ell'$, we compute the cosine similarity between $\mathbf{h}_i^{(\ell')}$ and every $\mathbf{r}_{j,t}^{(\ell)} \in \mathcal{R}$, obtain the top-$k$ reference vectors with the highest similarity, and return their corresponding sequences as descriptions. We note that, in contrast to EmbeddingLens and LogitLens, LatentLens uses several reference vectors per description, i.e., one for each token of the description and layer of the model. Therefore $\ell$ and $\ell'$ are not necessarily equal and a top-$k$ reference vector can come from a *different* layer than the visual token under inspection.

Following the example in Figure 2, a visual token representation $\mathbf{h}_i^{(\ell')}$ may have a nearest neighbor $\mathbf{r}_{j,t}^{(\ell)}$, of the contextualized representation of the token clocks in the sentence: "stone tower with gold clocks". Compared to EmbeddingLens and LogitLens, LatentLens can naturally be applied at every layer of a model and provides full sentence descriptions, allowing for more fine-grained interpretations.[4]

### 3.3. Evaluating interpretability

Determining whether a visual token representation is interpretable requires careful consideration of what is a semantic match between the description provided by an interpretability method and the underlying image patch. This is a complex task, where semantic matches can take many different surface forms.

We use GPT-5 (Hurst et al., 2024) as a judge to automate this process. Given an image with a red bounding box highlight-

ing the visual token region (cf. Figure 1) plus the 8 surrounding visual tokens, as well as the top-5 descriptions returned by either EmbeddingLens, LogitLens, or LatentLens, we prompt the judge to determine whether a description is interpretable, and to classify such cases as *concrete* (directly visible), *abstract* (conceptually related), or *global* (present elsewhere in image). A visual token is *interpretable* if at least one of the top-5 descriptions is classified as interpretable. We, the authors, validate this judge against human annotations across all three methods (EmbeddingLens, LatentLens, and LogitLens), totaling 1,020 instances. We find substantial agreement between the LLM judge and humans with Cohen's $\kappa = 0.68$. Judge prompt and details on human annotation are in Appendix C.

## 4. Experiments

Next, we describe our controlled setup of training projections between different vision encoders and LLMs (Section 4.1), interpret visual tokens with LatentLens first in our controlled setup (Section 4.2) and afterwards on off-the-shelf VLMs (Section 4.4). We also study to which LLM layers visual tokens align to (Section 4.3) and ablate how the size of the pool of contextual embeddings (Section 4.5).

### 4.1. Experimental setup

**Models and training.** Unless stated otherwise, we train the projection function between different combinations of vision encoders and LLMs using a controlled setup. We use three LLMs (OLMo-7B (Groeneveld et al., 2024), Qwen2-7B (Yang et al., 2024), LLaMA3-8B (Grattafiori et al., 2024)), and three vision encoders (CLIP-ViT-L/14-336 (Radford et al., 2021), DINOv2-L-336 (Oquab et al., 2024), and SigLIP-so400M-patch14-384 (Zhai et al., 2023)), resulting in a total of 9 model combinations. We note that compared to the other vision encoders, DINOv2-L-336 was pre-trained without any textual supervision. Models are trained following Molmo using the PixMo-Cap dataset (Deitke et al., 2025), with captions averaging 167 words and 9 sentences each. The trained projector `proj` is a 3-layer MLP and all other weights remain frozen. To reduce confounding factors, we directly use the patchified image as the input to each model. All models are trained for 12K steps with an effective batchsize of 8.

**Captioning performance.** We verify that our models produce reasonable captions using DCScore (Ye et al., 2025), a GPT-4o-based judge that rates captions on a 1–10 scale across fine-grained criteria (faithfulness, detail accuracy, hallucinations, completeness). Our models average 6.0/10, with CLIP and SigLIP encoders reaching an average of 6.8 while DINOv2 models score lower at avg. 4.4, likely due to DINOv2's lack of language supervision during pre-training. For reference, off-the-shelf Qwen2-VL-7B-Instruct (Wang

---

[4]While we focus on sentence-level, our approach can be trivially extended to phrases or even paragraphs.

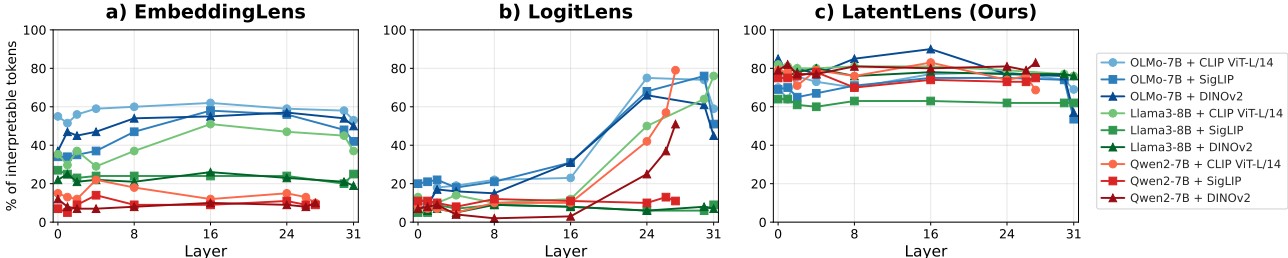

*Figure 3.* **Interpretability of visual tokens across layers using three different "lenses".** Each curve shows the percentage of interpretable visual tokens per layer across model. **(a)** EmbeddingLens: a large number of visual tokens is interpretable for OLMo variants but less for Llama3 and Qwen2. **(b)** LogitLens: low interpretability at early layers with a stark increase at later layers for most models. **(c)** LatentLens: the majority of visual tokens are interpretable across all models and layers.

et al., 2024) achieves a score of 8.5/10. See Appendix M.

**LatentLens setup.** We use 2.99M Visual Genome captions (Krishna et al., 2017) as our corpus $\mathcal{C}$ of descriptions. All captions are encoded by each individual LLM, and we store all contextual token representations for layers $\ell \in \{1, 2, 4, 8, 16, 24, L\text{-}2, L\text{-}1\}$ as reference vectors.[5] This encoding is a one-time cost per LLM backbone: the forward pass through ∼3M sentences takes ∼2h of GPU compute, with ∼13h total wall time including index building and ∼26 GB of storage across 8 layers (float8). Once the index is loaded, nearest-neighbor search takes ∼29 ms per image. We show in Section 4.5 that using just 1% of the corpus yields comparably strong interpretability while reducing storage to ∼250 MB.

**LLM-judge setup.** Although LatentLens retrieves full sentence descriptions, we only provide the *full* words corresponding to the top-5 contextualized token representations as descriptions for the LLM judge. We make this decision for two reasons: 1) We found that, in contrast to human annotators, the LLM-judge can get distracted by the sentence-level context, giving inconsistent results. 2) This setup provides a fair comparison to EmbeddingLens and LogitLens (both only return individual tokens as descriptions) by relying on the same exact judge prompt across all methods. We note that for LatentLens, this approach may even underestimate interpretability and we further investigate the benefit of sentence-level descriptions in Section 5.

### 4.2. Visual token representations are consistently interpretable across layers

The main question we investigate is how interpretable are visual token representations as they are processed by LLMs? To answer this question, we randomly sample 100 image patches from 100 images[6] from the PixMo-Cap validation

set and compare the fraction of interpretable representations for LatentLens, EmbeddingLens, and LogitLens[7] using the LLM judge described in §4.1.

**Results.** Figure 3 shows the fraction of interpretable visual token representations according to LatentLens, EmbeddingLens, and LogitLens for all models and layers. For EmbeddingLens (3a), we find that a reasonably large number of visual tokens are interpretable for some models, e.g., for all OLMo-based variants, 34–62% of visual tokens are interpretable from the input layer onward. For Qwen2-based models, however, only a small fraction of visual tokens are interpretable across layers (less than 20%). Llama3-based models fall in between with 20–50% of the visual tokens being labelled as interpretable.

For LogitLens (3b), we find that for all models, less than 25% of visual token representations are labelled as interpretable at lower layers. For all models except Llama3 + DINOv2, Llama3 + SigLIP, and Qwen2 + SigLIP much more visual token representations are interpretable at the later layers, e.g., 60–80% for all OLMo-based models at layers 24 and 30. This is consistent with the limitation of LogitLens discussed in Section 3, in particular its applicability only for layers close to the output layer. We additionally compare to Tuned Lens (Belrose et al., 2023), which learns per-layer affine probes on top of the LogitLens decoding step and find it does not improve visual token interpretability (see Appendix E for details).

In contrast, using LatentLens (3c) results in *consistently higher interpretability scores* across all layers. Moreover, there are only minor differences in interpretability scores across model combinations: almost all models have interpretable visual tokens in the range of 60–85% across all layers, with some OLMo variants dropping to around 53% at the final layer. We perform a series of ablations (see Appendix D for details), confirming that our findings are

---

[5]Due to memory constraints we store at most 20 different contextual representations per token in the model's vocabulary.

[6]To lower API costs, we test 100 patches, resulting in 3 methods × 100 patches × 9 models × 9 layers = 24.3K calls.

[7]In Appendix E we also study an improved version of LogitLens, TunedLens (Belrose et al., 2023), and find that it does not change our LogitLens results.

not tied to our specific training setup. We find no substantial change in LatentLens interpretability when reducing the expressivity of the mapping to linear, or when training on much shorter captions.

Overall, our results indicate that previous methods underestimate the interpretability of visual tokens and demonstrate the importance of using the right lens for analyzing latent representations. It is particularly interesting that visual tokens from DINOv2, with no explicit linguistic supervision, show consistently high interpretability with all three lenses.[8] Returning to our research question of whether visual tokens appear like interpretable words to the LLM, we can now answer: While visual token representations are not necessarily mapped one-to-one to the vocabulary of the LLM, they are often similar to contextual token representations which are semantically related to the image contents.

### 4.3. Mid-Layer Leap: Visual token representations tend to align to later layer text representations

LatentLens allows us to compare visual token representations at layer $\ell$ to contextual token representations from *any* LLM layer. We now investigate which latent textual representations are most similar to visual token representations. Given a visual token representation at layer $\ell$, we obtain the top-5 contextualized token representations with the highest cosine similarity and report the layer these contextualized representations are obtained from.

**Results.** Figure 4 shows the results of this experiment for all 9 model combinations. Surprisingly, visual token representations from early layers, and even the input layer, tend to have higher cosine similarity not with contextualized token representations from the same layer but instead with those from later layers. For example, with OLMo-7B + SigLIP, for visual token representations at layer 0, the majority of the nearest neighbor contextualized representations are from layer 8 of the LLM. Only once we reach mid-layers such as layer 8, we see a diagonal pattern where visual token representations are closest to contextualized representations from the same layer. For other model combinations, results are even more surprising. For Qwen2-7B + SigLIP, the most similar representations for visual tokens from any layer are always from layer 16.

Overall, these results suggests that the visual token representations align the most with more contextualized LLM representations rather than lexical representations at the input level. We refer to this as the *Mid-Layer Leap* phenomenon and provide additional analyses of this finding in Appendices F and H, investigating how much visual token representations change across layers and whether rogue dimen-

[8]The ability of DINOv2 to predict linguistic attributes of visual concepts has recently been reported (Oneata et al., 2025).

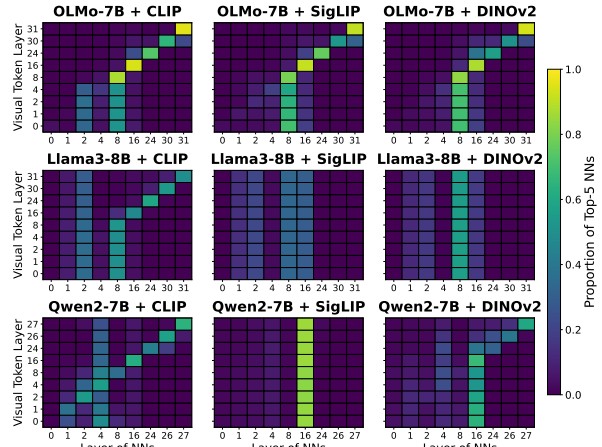

*Figure 4.* **The Mid-Layer Leap: early visual tokens align to later LLM layers.** For visual tokens at different stages of LLM processing, we compute their top-5 Nearest Neighbors from all other LLM layers. We find that early visual tokens, even at the input itself, align most to middle layers, e.g., layer 8 or 16. Some model combinations align most to a constant layer throughout processing, such as LLaMA3 variants. We analyze the L2 norm distributions and potential outlier effects in Appendix F.

sions (Timkey & van Schijndel, 2021) dominate the cosine similarity. We find that visual token representations change very little throughout layers, and no systematic evidence for rogue dimensions, respectively.

### 4.4. Results generalize to off-the-shelf VLMs

Finally, we validate that LatentLens generalizes beyond our controlled setup by applying all three lenses to six off-the-shelf VLMs spanning different scales: Molmo-7B-D and Molmo-72B (Deitke et al., 2025), LLaVA-1.5-7B (Liu et al., 2024a), LLaVA-NeXT-34B (Liu et al., 2024b), Qwen2-VL-7B-Instruct (Wang et al., 2024), and Qwen2.5-VL-32B-Instruct (Bai et al., 2025).

**Results.** Figure 5 shows interpretability across layers for all six models. LatentLens consistently achieves the highest interpretability on all six models. The Molmo models perform best (Molmo-7B-D: 86%, Molmo-72B: 78% average), consistent with their OLMo backbone being closest to our controlled setup. Qwen2-VL-7B-Instruct and LLaVA-1.5-7B reach 55–62% on average, while the larger Qwen2.5-VL-32B-Instruct and LLaVA-NeXT-34B are lower at 33–35%[9], though still substantially outperforming the baselines when averaged across all layers. EmbeddingLens and LogitLens typically reach 11–35% average interpretability, with the exception of EmbeddingLens on Molmo-72B (70%), also likely due to the OLMo backbone. We also verify that all six off-the-shelf models exhibit the same *Mid-Layer Leap* phe-

[9]It will be interesting for future work to explore why larger models have less consistent trends across layers

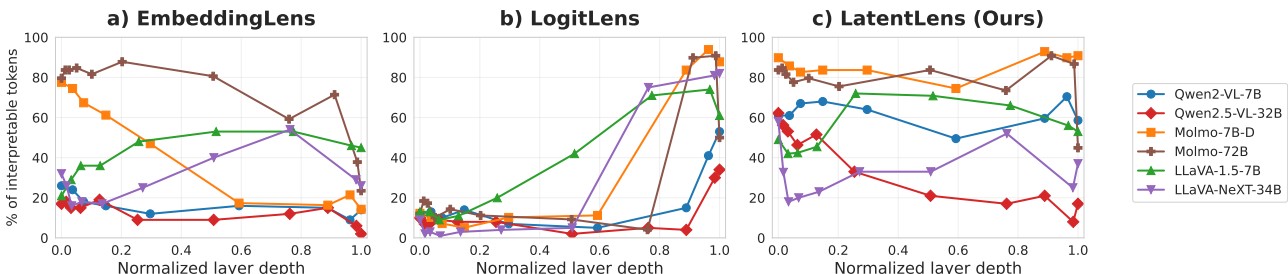

Figure 5. **Interpretability of visual tokens across six off-the-shelf VLMs (normalized layer depth).** We apply LatentLens and baselines to six off-the-shelf models spanning different architectures and scales. The x-axis shows normalized layer depth (0 = input, 1 = final layer) to enable direct comparison across models. LatentLens outperforms the baselines on all six models.

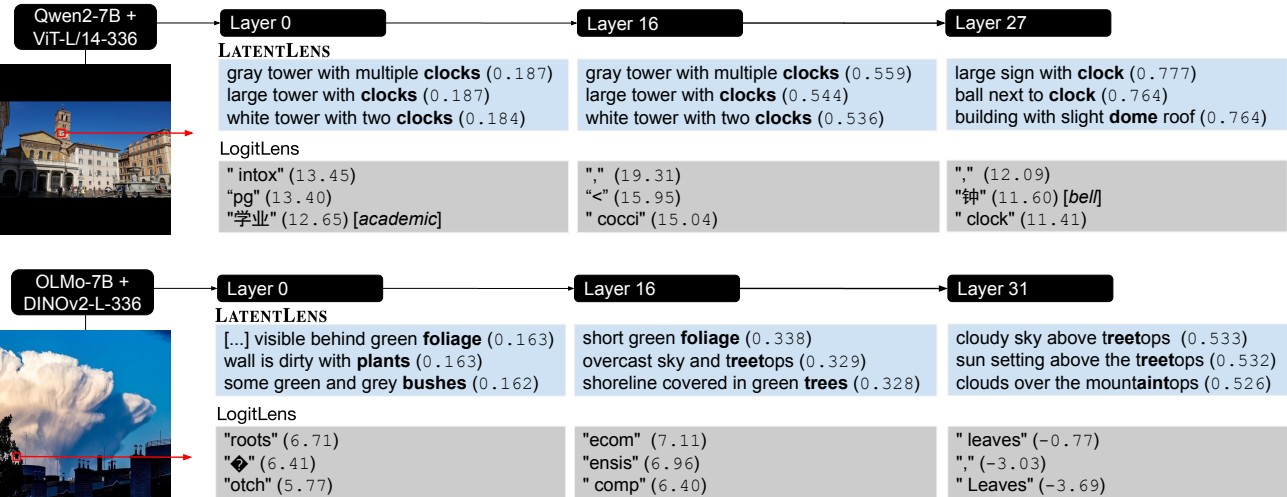

Figure 6. **Qualitative examples.** Highest-scoring descriptions are extracted from different combinations of LLMs and Vision Encoders using both LatentLens and LogitLens. The described patch is shown with a red outline, and the magnitude of scores are shown in parentheses. For LatentLens, the contextualized token used for the similarity calculation is shown in **bold**. Best viewed in colour.

nomenon observed in the controlled setup; see Appendix I for heatmaps.

### 4.5. Corpus size sensitivity

A natural question is how sensitive LatentLens is to the size of the corpus used to build the contextual embedding index. We ablate this by randomly sub-sampling from the full contextual embedding pool at rates of 0.1%, 1% and 10%. This is done across three model pairs (OLMo+CLIP, LLaMA3+SigLIP, Qwen2+DINOv2), measuring average interpretability across representative layers (0, 8, 16, 31). As shown in Table 1, interpretability drops meaningfully only below 1% (~58% avg.), while the difference between 1% and the full corpus is small in practice (67% vs. 72% on average). Intuitively, a smaller corpus does not change *whether* a token is interpretable but it reduces the precision and granularity of the descriptions, since fewer sentence contexts are available to match against.

Table 1. **Corpus size sensitivity.** Average % of interpretable visual tokens (LatentLens) for three model pairs and layers 0, 8, 16, 31. The difference between 1% and the full corpus is small.

| Corpus size | Avg. % interpretable |
|---|---|
| 0.1% | 58.4 |
| 1% | 67.3 |
| 10% | 72.5 |
| 100% | 71.6 |

## 5. Qualitative results

The previous section showed that under the right lens, visual token representations are highly interpretable. Here, we provide qualitative examples for the interpretability provided by LatentLens, and compare to LogitLens.

**LatentLens vs. LogitLens descriptions.** Figure 6 shows qualitative examples of the highest-scoring descriptions for two image patches extracted using LatentLens and LogitLens (see Appendix N and our interactive demo for addi-

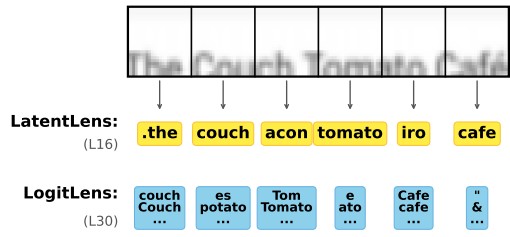

*Figure 7.* **LatentLens vs. LogitLens results on rendered text for OLMo + CLIP ViT-L/14-336.** LogitLens predicts plausible next tokens. We omit the surrounding sentence-level context for simplicity of visualization.

tional examples). LatentLens descriptions are semantically more meaningful than those of LogitLens. For example, even when LogitLens yields some interpretable tokens at a late layer on a patch of a church tower (first row), LatentLens provides highly accurate nearest neighbors across all layers such as "large tower with `clocks`". We also note that for LatentLens, the magnitude of the cosine similarity typically increases at deeper layers, i.e., the visual token representations become more similar to their textual nearest neighbors. For LogitLens, on the other hand, a higher similarity score (logit) does not necessarily translate into more interpretable descriptions. Furthermore, LogitLens descriptions often include unrenderable tokens, subwords, punctuation, and unrelated non-English tokens in Chinese. With LatentLens, however, we can simply merge the top matching token into a full word with adjacent subwords (since we have access to the entire sentence), if necessary, such as "b + `elf` + ry" into "belfry". Overall, these examples highlight the advantage of LatentLens: full-word and even sentence-level descriptions are more interpretable than subword tokens. We quantify this full-sentence advantage in Appendix K.

**Visually rendered text.** Images containing rendered text yield consistently interpretable results when using LatentLens. In Figure 7, we show an example for OLMo-7B + CLIP ViT-L/14-336. With LatentLens, the highest scoring contextualized representations correspond exactly to the words in the image.[10] When using LogitLens, on the other hand, we observe plausible next-token predictions which do not necessarily indicate what the visual token *intrinsically* encodes. For example, on the visual token where LatentLens predicts `couch`, LogitLens would predict `es` or `potato` instead. In other instances, LogitLens would even apply "correct" next-token-prediction on the rendered text and predict `Tomato`.

___
[10]For DINOv2, we would see generic words, e.g., "screenshot".

## 6. Related Work

We situate our work in the literature on understanding the mapping between vision and language models, the interpretability of VLMs, and analysing LLM representations.

**Connecting (frozen) vision and language models.** Several works have shown that LLMs can easily be adapted to process multimodal inputs (Tsimpoukelli et al., 2021; Mañas et al., 2023; Merullo et al., 2023; Lu et al., 2022), e.g., via small MLP or attention-based modules. Current state-of-the art VLMs (Yang et al., 2025; Deitke et al., 2025; Li et al., 2025) follow the same mapping paradigm but include more sophisticated image token pre-processing, adaptation training stages or connector designs. Some works, in spirit related to us, have also explored representing visual tokens explicitly as a weighted sum of the LLM vocabulary (Liao et al., 2025; Masry et al., 2025) or as discrete tokens (Chan et al., 2024), instead of a vanilla MLP projector. Various works study why an LLM can so easily adapt to non-linguistic inputs: Lu et al. (2022) frame LLMs as "universal computation engines" that can process any data sequence with minimal weight updates. Patel & Pavlick (2022) argue how LLMs, only trained on language, nonetheless learn implicit world models of the physical world, e.g., of color RGB spaces. Han et al. (2026) disentangle the visual priors learned during LLM's text-only pre-training into perception and reasoning priors. Our setup of understanding the frozen alignment between vision and language models is most similar to (Merullo et al., 2023), (Lu et al., 2022) and (Shukor & Cord, 2024), all of which explicitly keep the LLM frozen to characterize how vision integration is nonetheless possible.

**Interpreting VLMs.** Various interpretability methods, often initially developed for unimodal models (nostalgebraist, 2020; Huben et al., 2024; Belinkov, 2022), have been applied to VLMs (Neo et al., 2025), e.g., to characterize cross-modal concepts (Papadimitriou et al., 2025), cross-modality circuits (Nikankin et al., 2026), and the role of attention mechanisms in extracting information from visual tokens (Neo et al., 2025; Zhang et al., 2025). Most closely related to us are works that ask what a given visual token encodes in VLMs. Aside from training-based approaches like probes on visual tokens (Fu et al., 2025), many approaches leverage the inherent embedding space of the LLM to interpret visual tokens: E.g., Mokady et al. (2021) and Jiang et al. (2025a) study whether the LLM embedding matrix can interpret visual tokens, but only show qualitative examples on 1-2 models. LogitLens has been more widely used to interpret visual tokens at later LLM layers and applied to reduce hallucinations (Neo et al., 2025; Jiang et al., 2025b; Shukor & Cord, 2024; Park & Li, 2025; Wu et al., 2025). However, these works often involve either only small studies to quantify the interpretability via LogitLens, or only few models and with a closed set of object classes as labels. Overall,

neither of these works compare the interpretability via the embedding and unembedding matrix (Logitlens). To the best of our knowledge, no prior work has considered contextual embeddings for this purpose. Notably, Phukan et al. (2025) leverage the *average* intermediate contextual embedding of the generated answer to mitigate hallucination on VQA. They do not investigate the interpretability of visual tokens, and only rely on contextual embeddings from a single generated output (not a large collection). Past work also leverages a pool of text descriptions to interpret image tokens or contributions of model components in CLIP (Chen et al., 2024; Gandelsman et al., 2024).

Another fundamental question is how vision and language embedding spaces relate to each other, such as characterizing a fundamental "modality gap" (Liang et al., 2022; Jiang et al., 2024) or narrow-cone effects (Shukor & Cord, 2024) in models. Our findings can also broadly be seen as further evidence for a high structural similarity of vision and language representation spaces (Li et al., 2024), coined as the Platonic Representation Hypothesis (Huh et al., 2024).

**Representations in LLMs.** Prior work has investigated how other non-discrete tokens (i.e., soft prompts) or intermediate states are represented in LLMs. Soft prompts have been found to sometimes showcase interpretable neighbors, albeit generic concepts only broadly related to the task (Lester et al., 2021). Bailey et al. (2023) challenge this view of soft prompts as "word-like" units. Hidden representations and contextual word embeddings in language models have been extensively studied along axes such as layer evolution (Voita et al., 2019; Aken et al., 2020), embedding geometry (Ethayarajh, 2019) and semantics such as word sense disambiguation (Peters et al., 2018; Eyal et al., 2022; Wiedemann et al., 2019; Chang & Chen, 2019). Notably, Eyal et al. (2022) also leverage a large index of contextual word embeddings linked to the sentence they occurred in.

## 7. Discussion and Conclusion

In this work, we established that visual tokens are consistently interpretable across LLM layers, even at the input to the LLM. The ability to find interpretable visual tokens relies on using contextualized textual token representations, instead of the input or output embedding layers of the underlying model. Our findings challenge the inconclusive results of previous work in a systematic study across interpretability methods, models, and layers. These results also help explain why connecting separately trained vision encoders and LLMs requires only minimal weight and architecture updates. For example, one piece in the puzzle for explaining this straightforward mapping is our *Mid-Layer Leap* observation: visual tokens at the input are already aligned with semantic intermediate language representations (e.g., layers 8–16), rather than word-level embeddings.

Broadly, it is a long-standing goal to understand the relation between the physical world and abstract symbolic processing, both in human cognition (Harnad, 1990) as well as AI systems (Huh et al., 2024; Bisk et al., 2020). Our empirical study contributes toward understanding how visual and linguistic representations interface in neural systems, and to what extent we can find isomorphisms.

We foresee many exciting avenues for future work. A fundamental question is to what extent vision and language representations share deeper structural similarities beyond interpretability. LatentLens may extend beyond VLMs to other non-linguistic tokens such as soft prompts (Lester et al., 2021), latent thinking (Hao et al., 2025), or speech (Ògúnrèmí et al., 2025). It could also be applied to natively multimodal models (Team, 2024; Zhou et al., 2025; Team et al., 2023) or refined as a tool with a dynamic corpus.[11] In terms of downstream implications, we see promise in mitigating hallucination (Jiang et al., 2025b), as well as causal ablations investigating how interpretable vs. non-interpretable visual tokens affect task performance.

## Limitations

We acknowledge several limitations of our approach. First, LatentLens requires pre-computing and storing a large corpus of contextual embeddings, which incurs storage overhead compared to methods like LogitLens that rely solely on model weights. Second, our interpretability judgments may be influenced by the Visual Genome corpus of sentences used for contextual embeddings. Third, we crucially have not studied how or if certain LatentLens interpretations affect the downstream behavior of the VLM. Finally, while we study 15 VLM configurations, our findings may not generalize to architectures that differ substantially from the transformer-based models we analyze.

## Impact Statement

We contribute to the broader effort of understanding the internal representations of foundation models. Our work introduces interpretability tools that may help researchers and practitioners better understand how vision-language models process visual information. While it remains an open question whether interpretability methods will ultimately lead to more controllable and safer AI systems, we believe that fundamental research into model internals is a necessary first step toward this goal. More broadly, we hope that tools like LatentLens can help demystify these models for the wider public, moving away black-box perceptions and anthropomorphism toward a more grounded understanding of what these systems actually compute.

---

[11]We present an initial exploration of this in Appendix L

## Acknowledgments

We would like to thank our many colleagues from McGill NLP and the wider Mila community for their valuable feedback and brainstorming. BK is supported by the Vanier Canada Graduate Scholarships. DE was funded by IVADO Thematic Semester on Autonomous Agents and by research grant (VIL53122) from VILLUM FONDEN. MM is supported by the Mila P2v5 grant and the Mila-Samsung grant. SR is supported by Canada CIFAR AI Chairs program and IVADO R3AI. Finally, we thank Sonia Joseph and Elinor Poole-Dayan for final feedback on the draft.

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

## A. Limitations

We acknowledge several limitations of our approach:

- **Storage requirements:** LatentLens requires pre-computing and storing a large corpus of contextual embeddings, which incurs storage overhead compared to methods like LogitLens that rely solely on model weights. Our current corpus uses float8 compression to reduce storage to approximately 25% of float32 size, but the storage cost scales with both corpus size and the number of LLM layers analyzed.

- **Corpus constraints:** Our interpretability judgments are constrained by the Visual Genome corpus used for contextual embeddings. Domains not well-represented in this corpus (e.g., specialized scientific imagery, non-Western cultural contexts) may yield less meaningful interpretations. Future work could explore domain-specific corpora or dynamic corpus generation.

- **Noun bias:** The dominance of nouns in our nearest-neighbor results (approximately 45–50% as shown in Appendix J.2) may partly reflect corpus biases in Visual Genome rather than fundamental properties of visual token representations. Visual Genome's region descriptions naturally emphasize objects and entities over actions or relations.

- **Model scope:** While we study 15 VLM configurations (9 controlled setups plus 6 off-the-shelf VLMs), our findings may not generalize to architectures that differ substantially from the transformer-based models we analyze. In particular, we focus on frozen LLMs with MLP connectors; models with different connector architectures (e.g., Q-Former, Perceiver) or training paradigms may exhibit different interpretability patterns.

- **Evaluation subjectivity:** Despite validating our LLM judge against human annotations ($\kappa = 0.68$), interpretability judgments retain inherent subjectivity. What constitutes a "meaningful" interpretation depends on the task and context. Our binary interpretable/non-interpretable classification may miss nuances in interpretation quality.

## Appendix Overview

| | | |
|---|---|---|
| **A** | **Limitations** — Storage, corpus constraints, noun bias, model scope, evaluation subjectivity | |
| **B** | **LatentLens Design** — How the corpus of contextual embeddings was built and stored | |
| **C** | **LLM Judge** — LLM judge prompt, human annotation protocol, and validation | |
| **D** | **Ablations** — Robustness to seed, connector depth, caption detail, and task type; training dynamics showing when interpretability emerges | |
| **E** | **Tuned Lens Comparison** — Learned affine probes do not improve over LogitLens on visual tokens; LatentLens wins by 47–71 pp | |
| **F** | **L2 Norm Analysis** — Visual tokens have higher L2 norms and behave differently | |
| **G** | **Cosine Similarity Distributions** — Bimodal patterns in NN similarity | |
| **H** | **Layer Alignment Details** — Token drift through LLM processing | |
| **I** | **Off-the-Shelf Model Analysis** — Mid-Layer Leap heatmaps for all 6 off-the-shelf VLMs; deeper analysis for Qwen2-VL | |
| **J** | **Fine-grained Interpretation Analysis** — POS tags, visual attributes, interpretation types | |
| **K** | **Phrase-Level Interpretations** — How helpful phrase-level interpretations are compared to single words | |
| **L** | **Dynamic Corpus Generation** — Evolutionary search to generate better descriptions with maximal cosine similarity | |
| **M** | **Captioning Quality** — Ensuring our models produce good captions with DCScore | |
| **N** | **Qualitative Examples** — 20 non-cherry-picked for LatentLens, LogitLens and EmbeddingLens | |
| **O** | **Behind The Scenes** — Beyond the polished final product (this paper), we show stories, lessons learned, the messy reality of science | |

# B. LatentLens Design

## B.1. Contextual Embedding Corpus

For LatentLens, we extract embeddings from Visual Genome (Krishna et al., 2017) phrase-region annotations. We process 2.99M phrases (2,991,848 unique after deduplication) through each LLM, storing up to 20 contextual embeddings per vocabulary token at layers [1, 2, 4, 8, 16, 24, N-2, N-1] where N is the number of layers. This results in approximately 2.5M embeddings across 26,862 unique tokens per layer. To reduce storage requirements, embeddings are stored in float8 format (25% of fp32 size). We provide embeddings for OLMo-7B, LLaMA3-8B, Qwen2-7B, and Qwen2-VL-7B-Instruct.

The corpus was chosen because Visual Genome contains the right level of detailed descriptions for visual scenes and objects (e.g., "the door of a pickup truck", "multiple cows standing in a field"), providing contextual embeddings that are semantically relevant to our task of interpreting visual tokens. Each phrase is processed through the LLM, and we use reservoir sampling to limit storage to 20 embeddings per token per layer.

# C. Human Annotations and LLM Judge Design

We developed the LLM judge through iterative prompt refinement, particularly regarding: showing only the full image with a red box around the region of interest vs. additionally showing that cropped region as a separate image; how to explain the exact task in prompt format. We designed both a word-level and sentence-level judge, and find that the word-level is more reliable. Providing full sentences instead leads to more frequent over- or under-interpretations with LLM reasoning that finds associations where none might exist.

We then validate the LLM judge via correlation with humans: Do humans and the LLM judge mostly agree whether certain top-5 nearest neighbors are interpretable given a region in the image and the whole image context?

## C.1. LLM Judge Prompt

We use GPT-5 with the following prompt:

---

*You are a visual interpretation expert specializing in connecting textual concepts to specific image regions. Your task is to analyze a list of candidate words and determine how strongly each one relates to the content of the highlighted region.*

**Inputs**
1. **Full Image**: An image containing a red bounding box highlighting the region of interest.
2. **Cropped Region**: A close-up view of the exact region highlighted by the red bounding box. Only rely on this if it is too small in the full image (e.g. text is too small to read), otherwise rely on the full image.
3. **Candidate Words**: A list of words to evaluate.

**Evaluation Guidelines**
There are three types of relationships to consider between the candidate words and the highlighted region:

**Concrete**: A word is concretely related if it names something that is literally visible in the cropped region. This includes: objects or parts of objects clearly present; colors, textures, or materials visible; text, numbers, or symbols shown; shapes, patterns, or visual features.

**Abstract**: A word is abstractly related if it describes broader concepts, emotions, or activities related to what's shown in the cropped region, but isn't literally present. This includes: emotions or feelings (beautiful, scary, peaceful); activities or functions (driving, cooking, reading); cultural or conceptual associations (luxury, tradition, modern); qualities or characteristics (elegant, rustic, professional); anything deemed semantically related to the region or the whole image context.

**Global**: A word is globally related if it describes something that exists elsewhere in the full image (outside the highlighted region), but not in the cropped region itself. This includes: objects visible in other parts of the image; colors present in other parts; text or elements in different regions.

**Important Note:** For regions with text, the connection can be concrete (characters/words shown) or abstract (concepts implied by the text).

**Output Format**
Return a JSON object with: `reasoning` (string), `interpretable` (true/false), `concrete_words` (list), `abstract_words` (list), `global_words` (list).

---

*Figure 8.* Screenshot of the human annotation interface. Annotators see the image with a highlighted region (red box) and the top-5 candidate descriptions. For each candidate, they select whether it is related to the region concretely, abstractly, globally, or not at all.

### C.2. Human Annotation and Validation

To validate our automated LLM judge, the authors manually annotated visual tokens across all three interpretability methods: EmbeddingLens (360 instances), LatentLens (300 instances), and LogitLens (360 instances), totaling 1,020 annotations. For each instance, we showed annotators the image with the highlighted region and the top-5 candidate descriptions from the respective method (Figure 8). Annotators indicated which candidates (if any) were related to the region—either *concretely* (directly visible), *abstractly* (conceptually related), or *globally* (present elsewhere in image). Each instance was annotated by at least one author, with an average of 1.8 annotators per instance.

A visual token is labeled *interpretable* if a majority of annotators selected at least one candidate as related. Comparing human majority vote against the LLM judge, we find substantial agreement: Cohen's $\kappa = 0.68$ and accuracy = 84.4% across all 1,020 instances.

## D. Ablations

In this subsection we ablate several components of our training setup to determine to what extent our findings depend on our particular setup. In other words, will a given image patch be mapped to the same nearest-neighbour words, regardless of training data, task, or mapping function? Specifically, if we do find similar LatentLens interpretations of the same input across ablations, it could imply that visual tokens represent something akin to raw task-independent input and all the task-specific extraction and computation is conducted by the LLM.

**Experimental setup.** We focus on OLMo + CLIP-ViT and ablate the following 5 aspects:

1. The **random seed** used for initializing the connector weights and controlling the training data sampling.

2. Using a **linear mapping** instead of 3-layer MLP to map visual tokens into the LLM prefix space.

3. Varying the **level of detail** in the captioning dataset by training on single-sentence captions instead of the detailed multi-sentence Pixmo-Cap dataset.

4. **Unfreezing the LLM** during training. When allowing the LLM weights to adapt to the task of processing and describing visual tokens, two outcomes are plausible: the proportion of interpretable tokens either drops or it rises. For the former, we can conceive that some of the LLM weights specialize to be a visual processing model (and not a language model),

*Table 2.* **Ablation results showing LatentLens interpretability change and nearest-neighbor overlap with baseline**, averaged across all layers. △ **Interp.**: Change in % interpretable (baseline = 72.3%). **Top-5 NN Overlap**: Average number of matching neighbors out of 5. Token = same subword; Phrase = same full caption. Captioning ablations maintain interpretability with partial overlap (∼2/5), suggesting multiple valid mappings exist.

| Model Variant | △ Interp. | Top-5 NN Overlap | |
| --- | --- | --- | --- |
| | | Token | Phrase |
| Baseline (OLMo + ViT-L) | 72.3% | — | — |
| Different seed | +0.4% | 2.5 | 2.2 |
| Linear connector | −0.2% | 2.1 | 2.0 |
| First-sentence captions | −2.5% | 1.8 | 1.5 |
| Unfrozen LLM | +5.5% | 1.9 | 1.5 |
| Spatial Task (frozen) | −34.2% | 0.0 | 0.0 |
| Spatial Task (unfrozen) | −30.2% | 0.0 | 0.0 |

with less training pressure to represent tokens as words a frozen LLM can process. On the other hand, we can also imagine that now the connector module simply became more expressive, being able to align any visual token to LLM embeddings, even if the initial structure of both embedding space was sometimes non-trivially different.

5. **Changing the task** from captioning to a spatial prediction task. The model is trained to answer the question "Where is [OBJECT]?" with either "top" or "bottom". The hypothesis we test is whether the language-based task of captioning leads to the observed alignment of visual tokens to the LLM embedding space.

For each ablation we measure two metrics, averaged across all layers: (1) **LatentLens Interpretability**: The percentage of visual tokens judged interpretable by GPT-5, reported as change from baseline (72.3%). (2) **LatentLens Overlap**: Average number of matching top-5 LatentLens nearest neighbors with the baseline model (out of 5). We report both *token-level* overlap (same subword in both top-5 sets) and *phrase-level* overlap (same full contextual phrase).

**Interpretability is robust to training variations but requires language-based objectives.** Results are summarized in Table 2. Simply changing the seed results in only around half of the top-5 LatentLens results to overlap with the original run. All of them are still highly similar concepts but it is some indication that even the seed can have some influence, albeit it is unclear how semantically meaningful it is. On the other hand, we find encouraging results when replacing the 3-layer MLP connector with a single matrix (linear layer) and when replacing highly detailed captions as training data with single-sentence captions: The level of interpretability is almost the same, and we still see a significant amount of overlap between the original top-5 nearest neighbors (2.1 and 1.8 out of 5, respectively). Thus, the relationship between vision encoder and LLM embeddings spaces can be linearly aligned in an interpretable manner and moreover, short captioning data is enough to do so[12]. Next, we observe a notable increase in the amount of interpretable visual tokens (+5.5%) when unfreezing the LLM during training. Thus, with this more "expressive connector" in the form of some LLM weights, the model can learn a more interpretable alignment. Finally, we observe that the language-based task of captioning is necessary for interpretable visual tokens. When instead training the model to generate a single token ("top" or "bottom") given a question about the location of an object, interpretability drops by around 30%, with zero overlap to the original LatentLens top-5 NNs. The tokens that are still marked as interpretable for this spatial task are the same few generic words such as "upper", "background", or "outside".

**Training Dynamics** We investigate *when* LatentLens interpretability emerges during connector training. We re-ran the OLMo-7B + CLIP-ViT connector from scratch, saving intermediate checkpoints at steps 100, 1000, and 6000 (in addition to the final step 12000), and evaluated LatentLens on each using our LLM judge.

Results are shown in Table 3. At step 100 (0.8K training examples), interpretability is near-random (∼12%) across all layers. By step 1000 (8K examples, ∼8% of training), we find an interesting difference across layers: the earliest LLM layers (0–1) already reach 67% interpretability, while deeper layers (2–31) remain at only 17%. By step 6000 (48K examples, half of training), all layers converge to near-final performance. This suggests that alignment first forms at the input of the LLM — where visual tokens enter the residual stream — and only later propagates to deeper representations.

---

[12]Follow-up work could explore what the limit of simplicity of the training data is for this interpretable alignment to appear.

*Table 3.* **Training dynamics of LatentLens interpretability (OLMo+CLIP-ViT).** Average % of interpretable visual tokens across 9 LLM layers, plus the split between early layers (0–1) and deeper layers (2–31). Interpretability emerges first in the earliest layers at step 1000, while all layers reach near-final performance by step 6000.

| Step | Examples | Avg | Layers 0–1 | Layers 2–31 |
|---|---|---|---|---|
| 100 | 0.8K | 11.7% | 13.3% | 11.3% |
| 1000 | 8K | 28.1% | **67.0%** | 17.0% |
| 6000 | 48K | 75.1% | 75.4% | 75.0% |
| 12000 | 96K | 71.3% | 71.0% | 71.4% |

## E. Tuned Lens Comparison

Belrose et al. (2023) propose *Tuned Lens*, which learns a lightweight per-layer residual affine probe to improve upon Log-itLens. Each probe transforms the intermediate representation $\mathbf{h}$ as $\hat{\mathbf{h}} = \mathbf{h} + \mathbf{W}\mathbf{h} + \mathbf{b}$ (where $\mathbf{W}$ is initialized to zero so the probe starts as an identity map), and the result is then decoded with the unembedding matrix (the same final step as Log-itLens). The probes are trained to minimize the KL divergence between the probe's output distribution and the final-layer output distribution: $\mathcal{L} = \mathrm{KL}(p_{\mathrm{final}} \parallel q_{\mathrm{probe}})$.

**Implementation.** We follow the Belrose et al. (2023) recipe as close as possible. We use SGD with Nesterov momentum (lr=0.1, momentum=0.9, weight_decay=$10^{-3}$), a linear learning-rate schedule, and gradient clipping (clip=1.0). The weight decay regularizes each probe toward the identity transformation. We train the probes on visual token positions only, using 2,000 PixMoCap images ($\approx$1.15M visual tokens total), one probe per layer. We select four model pairs: the three lowest-LogitLens-scoring pairs from our main evaluation (LLaMA3+SigLIP 7.1%, LLaMA3+DINOv2 7.2%, Qwen2+SigLIP 10.8%) and the highest (OLMo+CLIP 34.3%), spanning the full range of LogitLens performance.

*Table 4.* **Tuned Lens vs. LogitLens and LatentLens (average % interpretable across layers).** Tuned Lens provides little or no improvement over LogitLens on visual tokens, while the main paper shows that LatentLens outperforms both by a large margin on every model.

| Model pair | LogitLens | Tuned Lens | LatentLens |
|---|---|---|---|
| LLaMA3-8B + SigLIP | 7.1 | 8.9 | 62.3 |
| LLaMA3-8B + DINOv2 | 7.2 | 6.4 | 77.4 |
| Qwen2-7B + SigLIP | 10.8 | 7.4 | 74.3 |
| OLMo-7B + CLIP | 34.3 | 25.2 | 72.3 |

**Results.** Tuned Lens yields at best a marginal gain (+1.8 pp on LLaMA3+SigLIP) and is worse than LogitLens on the other three pairs. On OLMo+CLIP, the model where LogitLens already works reasonably well, early-layer probes do improve interpretability, but this benefit reverses at later layers (a collapse of 31–44 pp at layers 24–31), despite the identity-initialised probes and $\ell_2$ weight decay that were designed to prevent exactly this kind of drift. The result is a net decrease of $-9.1$ pp relative to plain LogitLens.

## F. Vision vs. Text Token L2 Norms

Figure 9 shows the L2 norm distributions for all 9 model combinations (3 LLMs $\times$ 3 vision encoders). For each model, we show separate histograms for visual tokens (left) and text tokens (right), colored by layer (yellow=early, red=late). Key observations:

- **Vision tokens have larger L2 norms than text tokens** across all models, often by 1–2 orders of magnitude.

- **OLMo-7B** maintains relatively small L2 norms (max $\approx$1000) for both modalities.

- **LLaMA3-8B and Qwen2-7B** exhibit much larger L2 norms for visual tokens, with max values exceeding 100,000 in some cases.

- **DINOv2** consistently produces the largest L2 norms across all LLMs.

- The 99th percentile (p99, black dotted line) and maximum (red dashed line) markers show substantial outliers in visual token distributions.

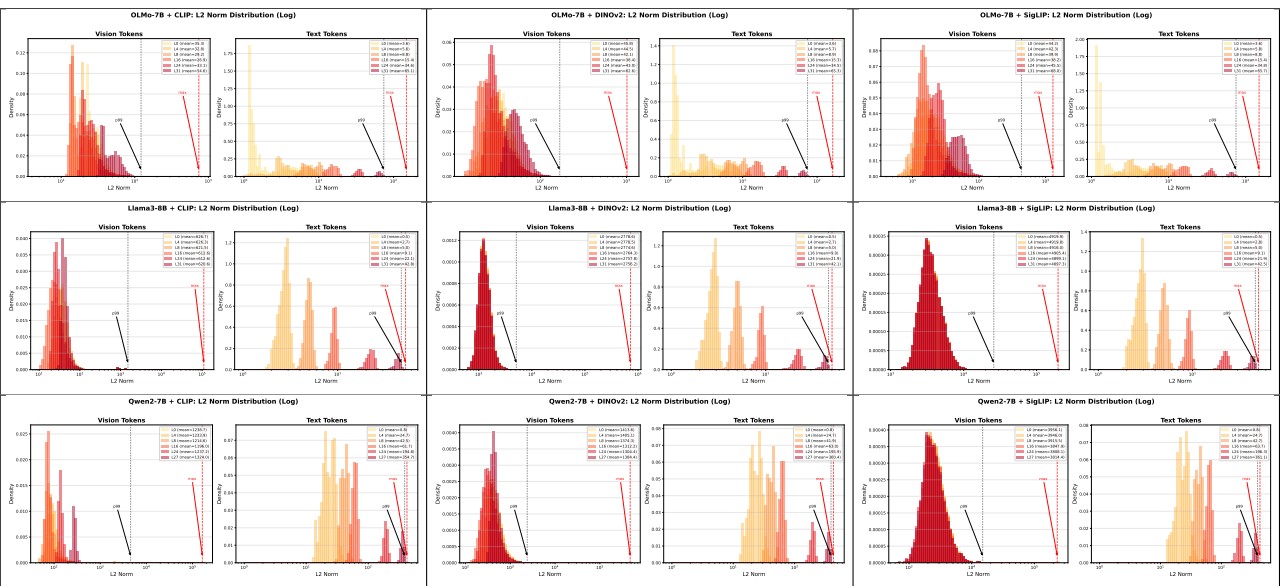

*Figure 9.* **L2 norm distributions of vision vs. text tokens across layers.** Each row corresponds to an LLM (OLMo-7B, LLaMA3-8B, Qwen2-7B), and each column shows a vision encoder (CLIP, DINOv2, SigLIP). Within each cell, Vision tokens (left) and Text tokens (right) are shown. Colors indicate layer depth (yellow=early, red=late). The x-axis uses log scale. Black dotted lines mark the 99th percentile; red dashed lines mark the maximum value.

**Are High L2 Norms from Sparse Outliers or Uniform Scaling?**    To understand whether high L2 norms are driven by a few large embedding dimensions (sparse outliers) or uniformly larger values across all dimensions, we extract the full embedding vector of the visual token with maximum L2 norm for each model and analyze its distribution.

Figure 10 shows histograms of individual embedding dimension values for these max-L2-norm tokens. The key finding is striking:

- **All distributions are approximately Gaussian**, not sparse. High L2 norms come from uniformly larger values across *all* 3584–4096 dimensions, not from a few extreme outliers.

- **OLMo-7B has ∼100× smaller embedding scale** than LLaMA3-8B and Qwen2-7B. Standard deviations are ∼10–20 for OLMo vs. ∼1700–11000 for LLaMA3/Qwen2.

- **LLaMA3-8B max tokens occur at layer 0** (input), while OLMo and Qwen2 max tokens occur at **layer 24** (late layers).

This suggests that the MLP connector (or already the vision encoder itself) learns fundamentally different embedding scales depending on the target LLM architecture, with LLaMA3 and Qwen2 resulting in much larger magnitude projections than OLMo.

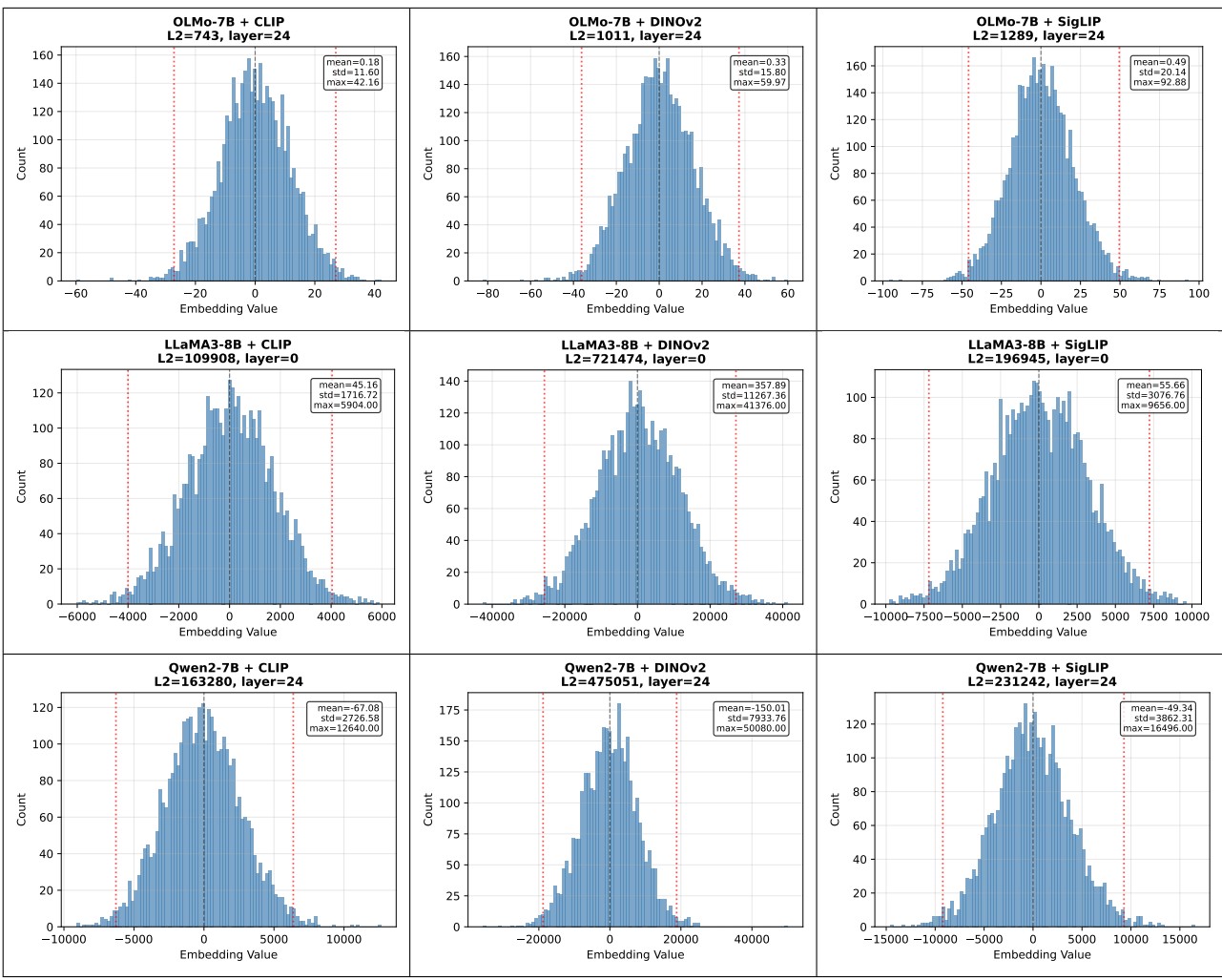

*Figure 10.* **Distribution of embedding dimension values for max L2 norm visual tokens.** Each row corresponds to an LLM (OLMo-7B, LLaMA3-8B, Qwen2-7B), and each column shows a vision encoder. We extract the visual token with the largest L2 norm and plot a histogram of its individual embedding dimension values. All distributions are Gaussian-like, indicating that high L2 norms result from uniform scaling across all dimensions rather than sparse outliers.

## G. Cosine Similarity Distributions

Figure 11 shows the distribution of top-1 nearest neighbor cosine similarities between visual tokens and their closest contextual text embeddings at layer 16 (a representative mid-layer). Several model combinations exhibit **bimodal distributions** with two distinct peaks. This pattern is most pronounced for LLaMA3-8B + CLIP and Qwen2-7B + CLIP, where one peak appears around 0.15–0.2 and a second around 0.35–0.4. A similar bimodal pattern is visible for Qwen2-7B + DINOv2.

In contrast models paired with SigLIP tend to show more unimodal distributions. The bimodal structure suggests that visual tokens may fall into two categories: those that align well with text representations (higher similarity) and those that remain more distant from the text embedding space (lower similarity). This could correspond to tokens representing salient visual content versus background or less semantically meaningful regions.

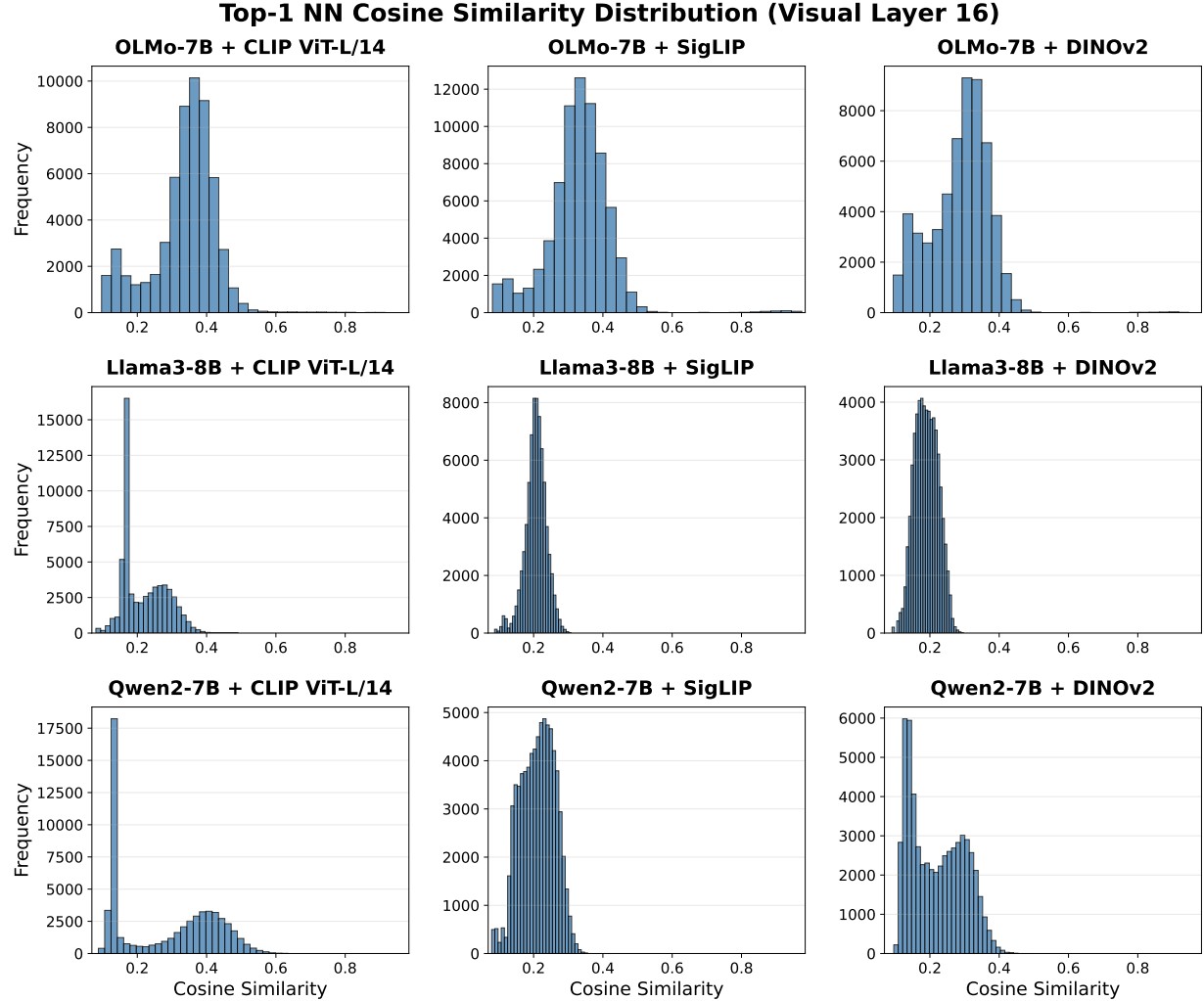

*Figure 11.* **Distribution of top-1 nearest neighbor cosine similarities at layer 16.** Each panel shows the histogram of cosine similarities between visual tokens and their closest contextual text embedding for one LLM–vision encoder combination. Several combinations (notably LLaMA3 + CLIP, Qwen2 + CLIP) exhibit bimodal distributions, suggesting distinct populations of visual tokens with different degrees of alignment to the text embedding space.

## H. Layer Alignment Details

To understand why visual tokens at the input layer align most strongly with contextual text representations from *later* layers (the Mid-Layer Leap, Section 4.3), we investigate how much visual token representations change as they pass through the LLM.

Figure 12 shows the cosine similarity between each token at layer $\ell$ and its own representation at layer 0, averaged across all tokens. For text tokens, similarity to the input embedding drops rapidly within the first few layers (often below 0.4 by layer 4), reflecting the contextualization process where tokens incorporate information from surrounding context. In contrast, visual tokens maintain much higher similarity to their input representations—often above 0.8 through mid-layers—indicating they undergo substantially less transformation during LLM processing.

This minimal drift of visual tokens explains the Mid-Layer Leap: since visual tokens are already "pre-contextualized" by the vision encoder and connector, they arrive at the LLM in a representational state more similar to how the LLM represents text after several layers of processing. The frozen LLM then processes these tokens with relatively little modification, as evidenced by the high layer-to-layer similarity.

## Same-Token Similarity Across LLM Layers

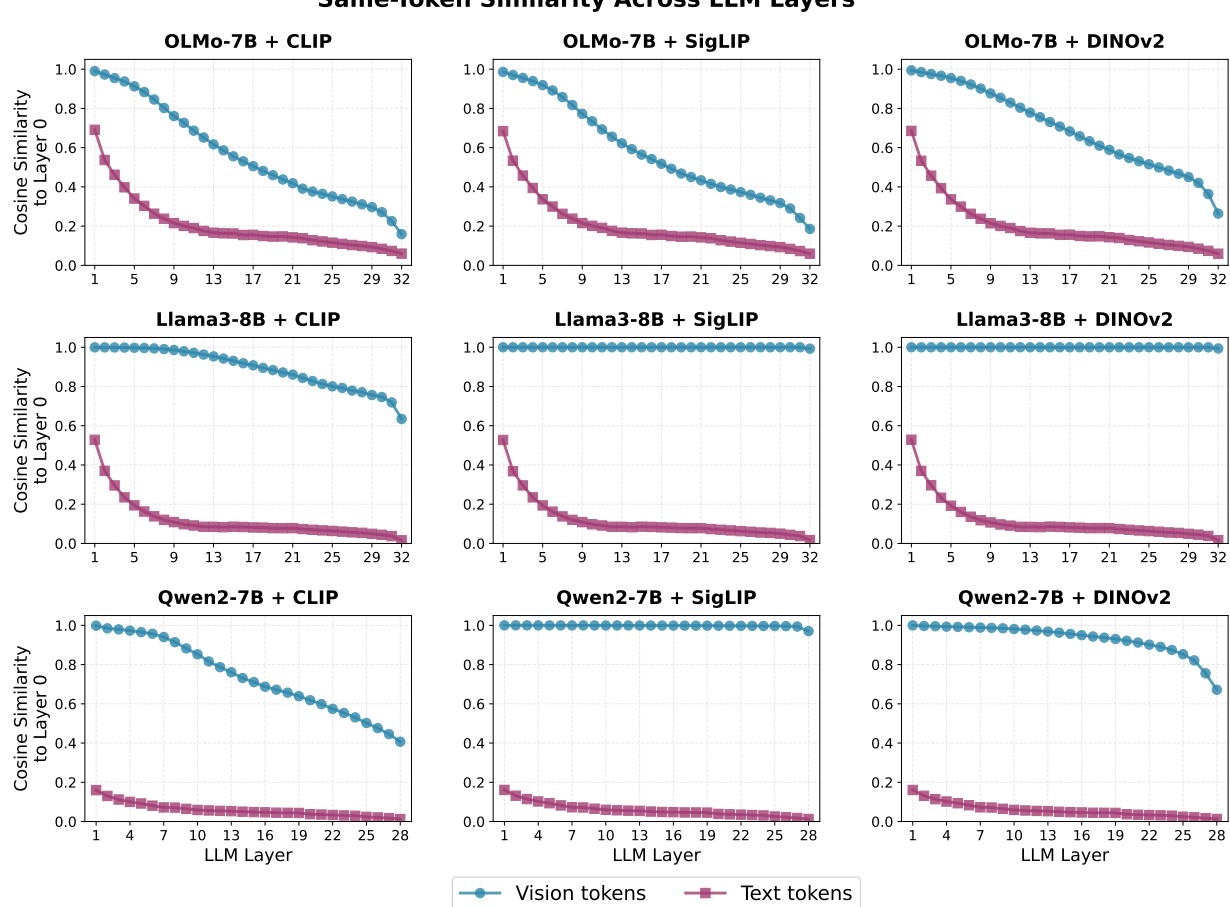

*Figure 12.* **Vision tokens follow only minor drift through LLM processing:** We compare the same token (e.g. same position in the image or text passage) to its input-layer embedding across layers. We find that text tokens early on have little similarity with their initial embeddings, perhaps following some process of contextualization, abstraction, or simply preparing for next-token prediction. On the other hand, visual tokens display a much higher cosine similarity to their input embeddings, especially until middle layers.

# I. Results for off-the-shelf VLMs

This section provides additional details for Section 4.4. We verify that the *Mid-Layer Leap* generalizes to all six off-the-shelf VLMs, and provide a deeper analysis for Qwen2-VL-7B-Instruct.

**Mid-Layer Leap across all six models**   Figure 13 shows the layer alignment heatmaps for all six off-the-shelf VLMs. All six exhibit the same hallmark pattern observed in the controlled setup (Figure 4): visual tokens at layer 0 align primarily to middle LLM layers (the leap), while from mid-processing onward a diagonal pattern emerges. This confirms that the Mid-Layer Leap is not an artifact of our controlled training setup but a robust property of how visual tokens interact with LLM representations.

**Deeper analysis: Qwen2-VL-7B-Instruct**   Among the six off-the-shelf models, we conduct a more detailed analysis for Qwen2-VL-7B-Instruct (Wang et al., 2024) as a representative example. Qwen2-VL differs from our controlled training along several axes: it was trained in multiple stages on 1.4 trillion tokens; crucially the LLM and vision encoder are unfrozen at some stages; it uses various different datasets at different stages, e.g. instruction tuning data; and finally it relies on elaborate image token preprocessing. We apply the same evaluation methodology as before: 100 random visual token patches evaluated by our LLM judge across layers $\ell \in \{0, 1, 2, 4, 8, 16, 24, 26, 27\}$. Since Qwen2-VL uses a finetuned version of Qwen2's LLM (not the frozen base model), we extract contextual embeddings from Qwen2-VL's own LLM backbone to ensure proper alignment.

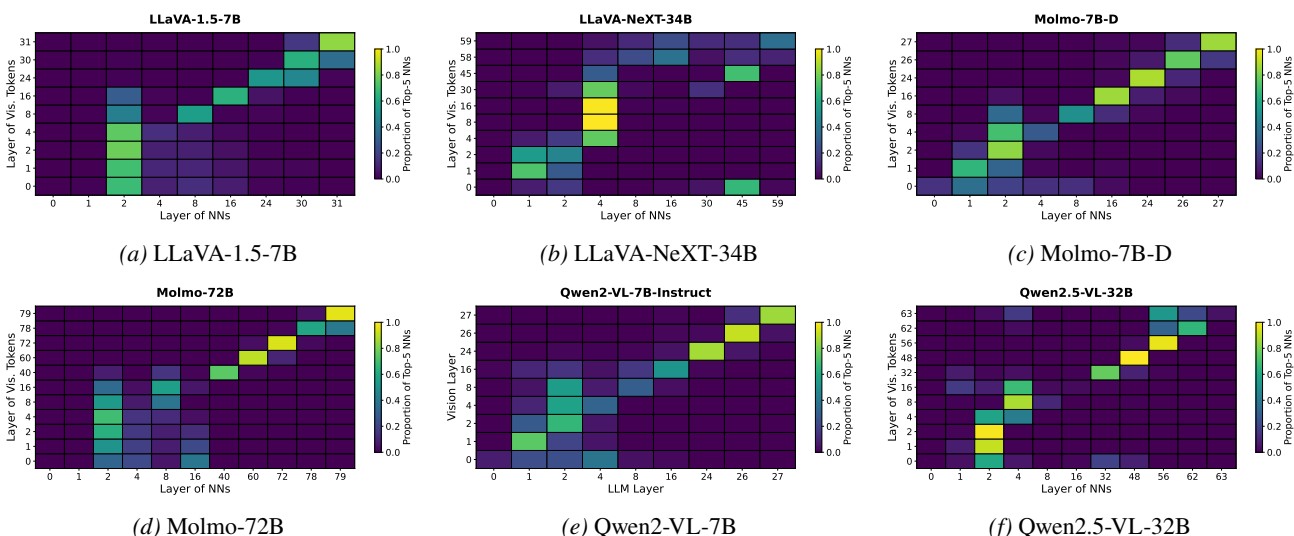

*Figure 13.* **Mid-Layer Leap in six off-the-shelf VLMs.** Layer alignment heatmaps showing which LLM layer's contextual embeddings are most similar to visual tokens at each processing stage. All six models show the same pattern as the controlled setup: a leap at layer 0 toward middle layers, then a diagonal alignment from mid-processing onward.

Figure 14 shows the layer alignment and token drift for Qwen2-VL. For visual tokens at layer 0, the leap is to LLM layer 4, then from layer 1 onward a clear diagonal pattern emerges. Visual tokens also undergo more change throughout LLM layers than in our frozen-LLM controlled setup, though still substantially less than text tokens (see Figure 12).

## J. Fine-grained Interpretation Analysis

Beyond measuring *whether* visual tokens are interpretable (Section 4), we analyze *what kinds* of interpretations LatentLens produces. We examine three dimensions: (1) interpretation types—whether nearest neighbors describe something concrete, abstract, or global (based on LLM judge outputs); (2) parts-of-speech—the grammatical categories of matched words (using spacy from Honnibal et al. (2020)); and (3) visual attributes—how often color, shape, and texture words appear. Key findings: concrete interpretations dominate (65–75%), nouns are most common (45–50%), and color words decline from early to late layers while other ratios remain stable—suggesting the frozen LLM does not progressively abstract away from concrete visual content.

### J.1. Interpretation Types

Figure 15 provides a visual breakdown of interpretation types aggregated across all models. Figure 16 shows the breakdown for all 9 model combinations individually. We categorize LatentLens interpretations as:

- **Concrete**: The nearest neighbor word directly names something visible in the image region (objects, colors, textures).

- **Abstract**: The word describes a concept related to but not literally visible (emotions, activities, functions).

- **Global**: The word describes something present elsewhere in the image but not in the highlighted region.

Across all models and layers, concrete interpretations dominate (70–75% on average), with abstract and global each contributing 11–15%. The ratios remain remarkably stable across layers, suggesting that the frozen LLM does not progressively "abstract away" from concrete visual descriptions.

As a representative off-the-shelf model, Figure 17 shows the same analysis for Qwen2-VL-7B-Instruct (see **??** for further details on this model). The pattern is similar: concrete interpretations dominate (62–78%), though we observe a slight decrease in later layers (26–27: 62%) with a corresponding increase in abstract interpretations. This may reflect Qwen2-VL's unfrozen LLM having learned some layer-wise abstraction.

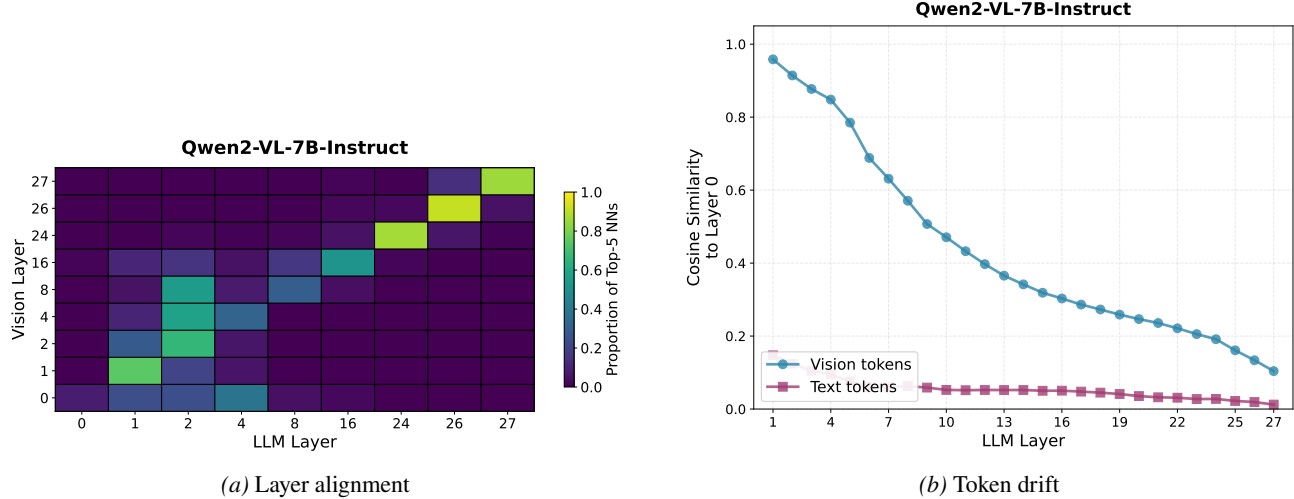

*Figure 14.* **Qwen2-VL layer analysis.** (a) Which LLM layer's contextual embeddings are most similar to visual tokens at each processing stage. (b) Cosine similarity of tokens at each layer to their input-layer representation. Visual tokens undergo substantial transformation (0.96→0.10), while text tokens start low ( 0.15) and stay low—unlike frozen LLMs where visual tokens retain higher similarity.

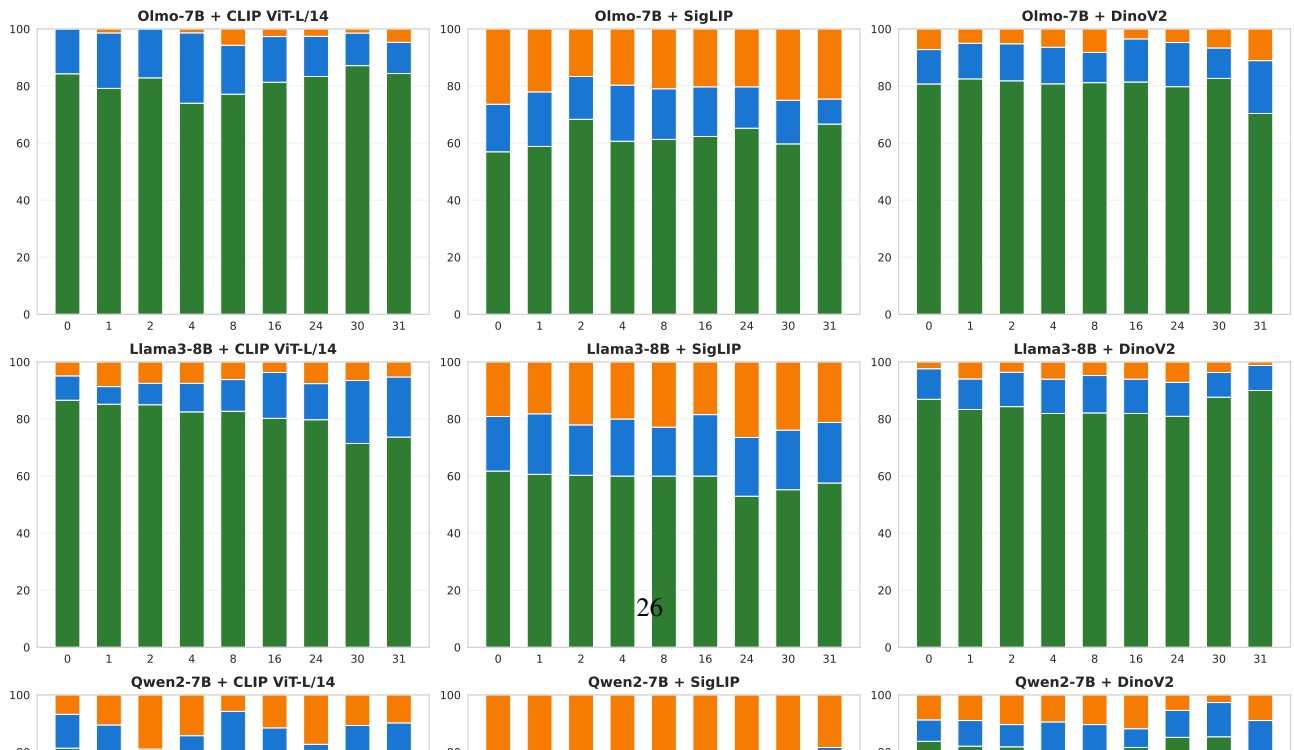

*Figure 15.* **Breakdown of interpretation types.** Top row: categories (Concrete 65%, Abstract 19%, Global 16%). Second row: top most frequent nearest-neighbor words per category. Lower rows: example Visual Genome phrases showing context, with target word in **bold**. Data aggregated across all 10 models (9 controlled setups plus Qwen2-VL).

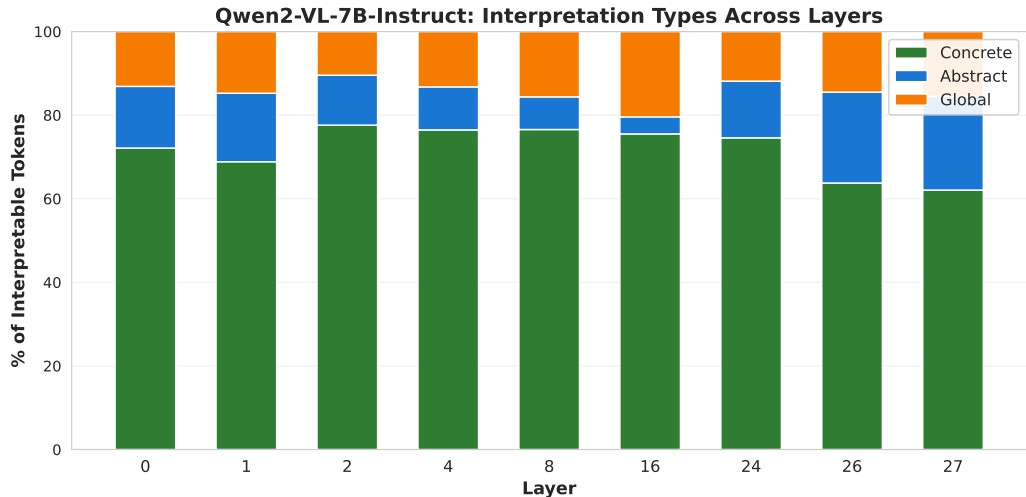

*Figure 17.* **Interpretation types for Qwen2-VL-7B-Instruct.** Similar to the frozen LLM models, concrete interpretations dominate. A slight decrease in concrete (and increase in abstract) is observed in later layers (26–27), possibly reflecting the unfrozen LLM's learned abstraction.

## J.2. Parts-of-Speech and Visual Attributes

Figure 18 shows the distribution of parts-of-speech among all nearest neighbors for each of the 9 trained model combinations. Nouns dominate at approximately 45–50%, which aligns with the visual nature of the interpretations—visual tokens primarily encode objects and entities. Proper nouns account for 10–20%, verbs 10–15%, and adjectives around 5%. The distribution is relatively stable across layers, with some variation between models (e.g., OLMo-7B + ViT-L shows more variability). Figure 19 shows the same analysis for Qwen2-VL-7B-Instruct, which exhibits similar patterns.

Figure 20 shows the frequency of visual attribute words (colors, shapes, textures) for each trained model. Color words are the most common at around 5–6% in early layers, declining to around 3% in later layers. This suggests that raw color information is more prominent in early visual token representations. Shape and texture words are rare throughout (<1%). Figure 21 shows the same for Qwen2-VL.

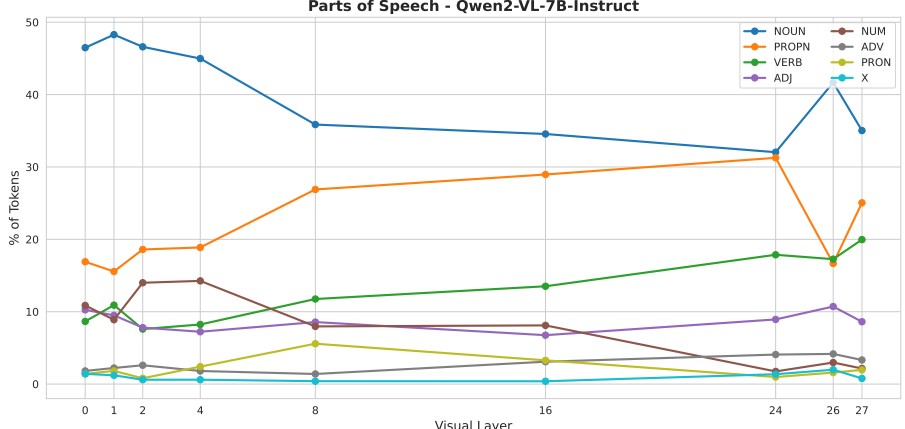

*Figure 18.* **Parts-of-speech distribution across layers for 9 trained models.** Nouns dominate (45–50%), followed by proper nouns and verbs. The distribution is relatively stable across layers with some per-model variation.

*Figure 19.* **Parts-of-speech distribution for Qwen2-VL-7B-Instruct.** Similar pattern to trained models: nouns dominate, followed by proper nouns and verbs.

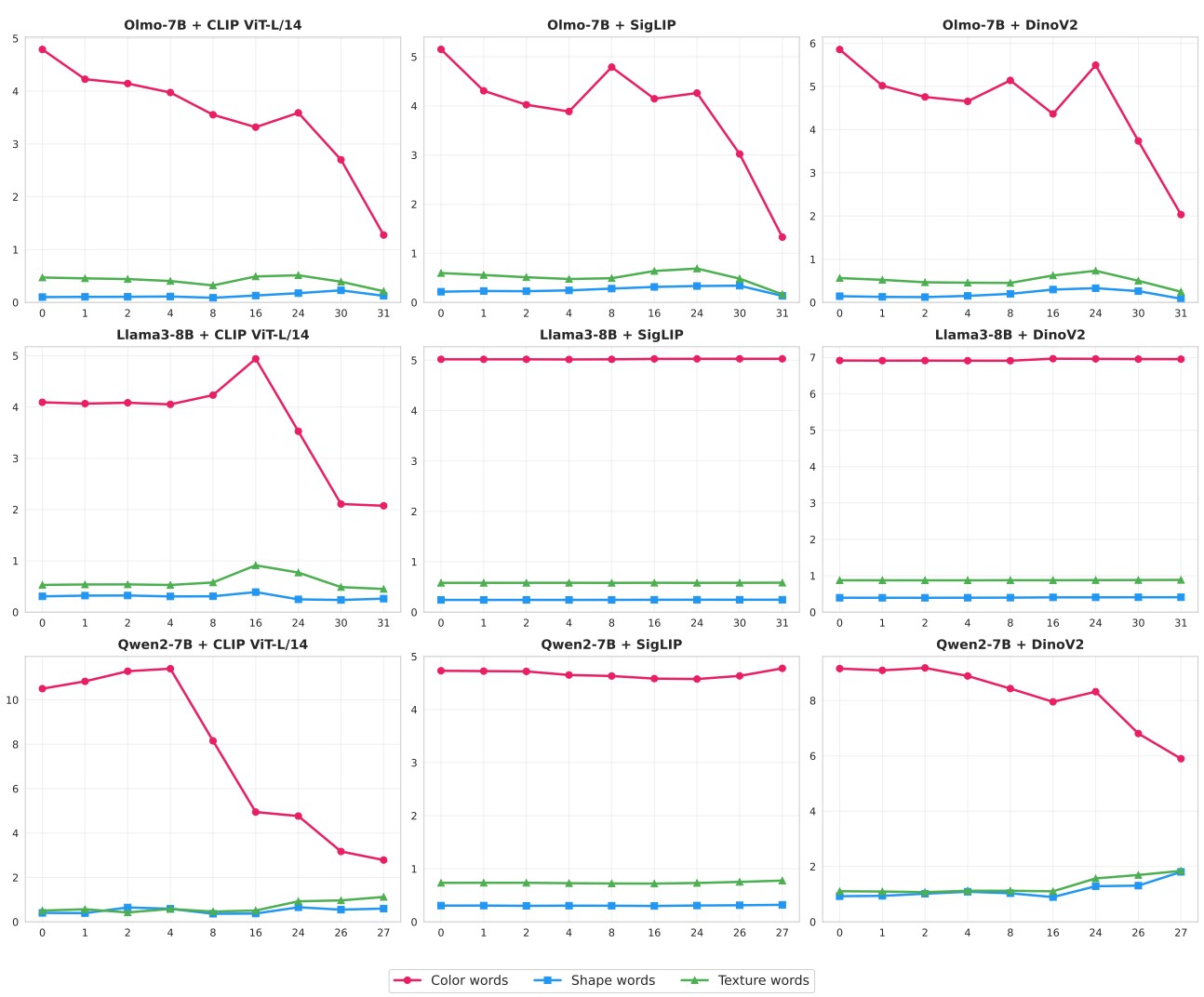

*Figure 20.* **Visual attribute word frequency for 9 trained models.** Color words are common (5–6% early, declining to 3% late), while shape and texture words are rare (<1%).

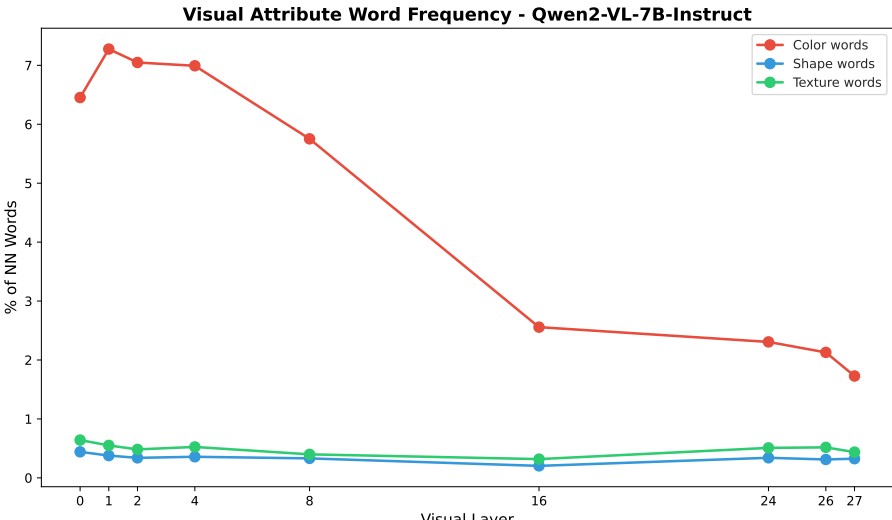

*Figure 21.* **Visual attribute word frequency for Qwen2-VL-7B-Instruct.** Similar pattern: color words dominate visual attributes, shape and texture are rare.

## K. Phrase-Level Interpretation Examples

**Quantifying the value of context.** To measure how much the preceding context contributes to interpretation quality, we randomly sample 50 examples where the LLM judge deemed the top LatentLens result interpretable. For each example, we manually label whether the preceding context helps interpretation compared to the word alone, using three categories: *yes* (context clearly aids understanding), *neutral* (context neither helps nor hurts), or *no* (context is misleading or irrelevant).

For instance, comparing "large stone tower with gold **clocks** " to just " **clocks** "—the former provides richer spatial and material context that better matches the visual content.

We find that in **64%** of cases, the preceding context provides a better interpretation than the word alone. In 28% of cases the context was neutral (the word alone was sufficient), and in only 8% was the context misleading. This validates that LatentLens's phrase-level interpretations offer meaningful advantages over token-level approaches.

**Qualitative examples.** We show 12 randomly selected (non-cherry-picked) examples from our phrase annotation study. Each panel shows the visual token's patch (red box) with preprocessing matching the respective vision encoder (CLIP preserves aspect ratio with padding, SigLIP and DINOv2 squash to square). Below each image we show: the LatentLens phrase versus a random Visual Genome phrase containing the same token. The highlighted word shows the matched token, demonstrating how contextual information can sometimes aid interpretation.

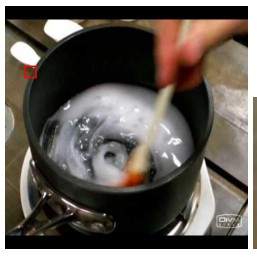 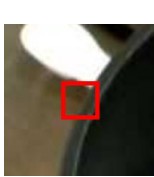

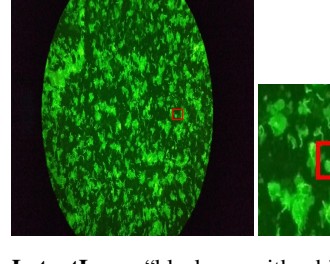

**LatentLens:** "pals : the pair were spotted posing in matching person - branded aprons at the event , with a large blue car parked **behind** them"

**Random phrase, same token:** "the trees are **behind** the giraffe"

*LLaMA3+DINOv2, L4*

**LatentLens:** "black eye with white **specks** around it"

**Random phrase, same token:** " **rocks** beside zebra"

*LLaMA3+SigLIP, L30*

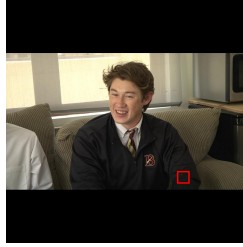 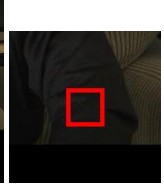

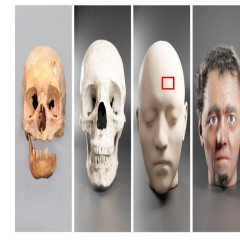 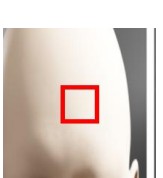

**LatentLens:** "woman wearing black sweatpants and grey **long-sleeve** t-shirt"

**Random phrase, same token:** "photo was taken by jenny **lee** silver"

*LLaMA3+CLIP, L16*

**LatentLens:** "the blue shirt the **bald** man has on."

**Random phrase, same token:** "smiling **bald** man"

*OLMo+DINOv2, L4*

*Figure 22.* **Phrase annotation examples (1–4).** Each panel shows a vision token's patch (red box) preprocessed as the vision encoder sees it. **LatentLens**: Top contextual nearest neighbor phrase from LatentLens. **Random**: A random VG phrase containing the same token.

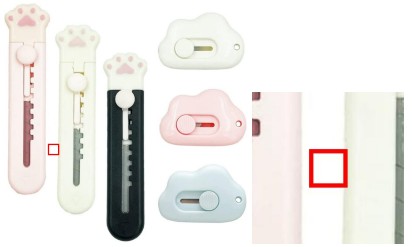

**LatentLens:** "multiple types of cherry tomatoes in multiple colours , all in bluegreen boxes"

**Random phrase, same token:** "something black w/ wild colours deflated before the tent; hot air balloon, maybe?"

*OLMo+SigLIP, L4*

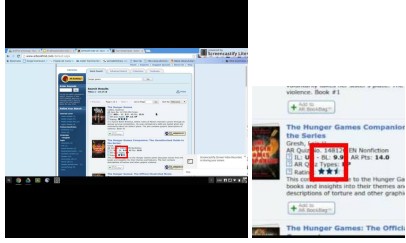

**LatentLens:** "five stars on the side of the bus"

**Random phrase, same token:** "the stars below the plane."

*OLMo+CLIP, L30*

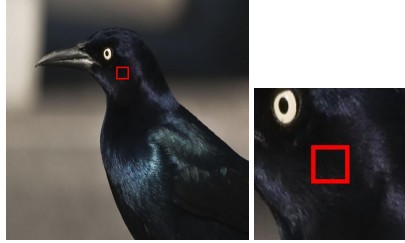

**LatentLens:** "the cow is mostly black "

**Random phrase, same token:** "the non black car on the roadway."

*Qwen2+DINOv2, L8*

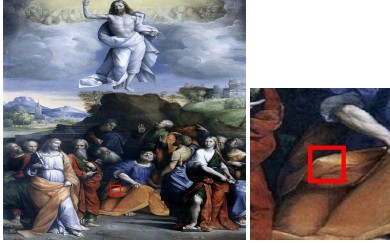

**LatentLens:** "man wearing white short sleeve tunic "

**Random phrase, same token:** "two unicorns dancing together"

*Qwen2+SigLIP, L0*

*Figure 23.* Phrase annotation examples (5–8).

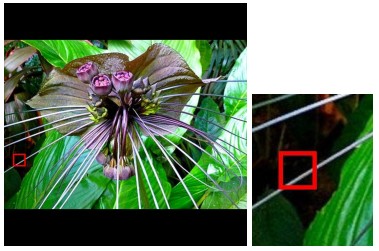

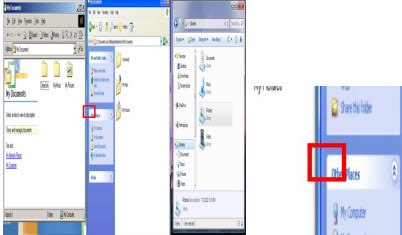

**LatentLens:** "green leaves are in the ==**background**== ."
**Random phrase, same token:** "the green vegetation in the ==**background**== "

*Qwen2+CLIP, L24*

**LatentLens:** "bowl is white and has two ==**sections**== "
**Random phrase, same token:** "dried splinted wood ==**sections**== "

*LLaMA3+DINOv2, L16*

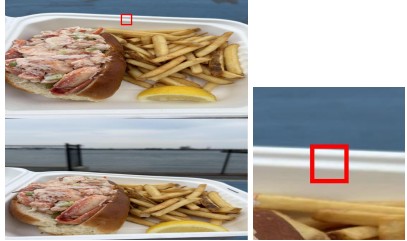

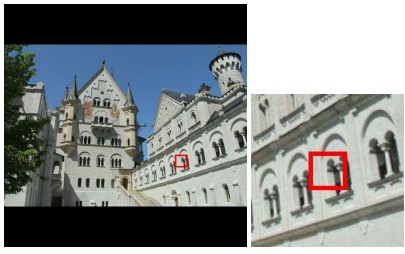

**LatentLens:** "barbecue sauce on the side of a ==**styrofoam**== cup"
**Random phrase, same token:** "a metal ==**rooster**== on a pole"

*LLaMA3+SigLIP, L1*

**LatentLens:** "a pale coloured wooden model of an old house with pitched roof and gable ==**windows**== stands on a table ."
**Random phrase, same token:** "row of three ==**windows**== "

*LLaMA3+CLIP, L16*

*Figure 24.* Phrase annotation examples (9–12).

## L. Dynamic Corpus Generation

As noted in Section 5, our fixed corpus contains at most 20 contextual embeddings per vocabulary token. We investigate whether *dynamically generating* phrase contexts—rather than relying on a fixed corpus—can yield better LatentLens interpretations.

**Method.** We use an evolutionary search approach: given a visual token and its top-5 LatentLens nearest neighbors from the fixed corpus, we iteratively generate variations using GPT-4o and keep those with highest cosine similarity to the visual embedding. Crucially, since LLMs are autoregressive, we only modify words *before* the target token (the token whose contextual embedding we extract), keeping the target token at the end of each phrase.

Specifically, we run 6 rounds of evolution with 20 variations per round, keeping the top-5 phrases. We evaluate on 20 visual tokens that our LLM judge marked as interpretable (OLMo-7B + CLIP ViT-L/14, layer 16).

**Results.** Dynamic generation improves cosine similarity in 85% of cases (17/20), with an average improvement of +0.017. Table 5 shows representative examples. The evolved phrases tend to be more concise and visually specific—for instance, "a white and black peaked **building** with a peaked roof" (similarity 0.415) evolves to "grand arched beige **building**" (similarity 0.463).

Interestingly, in 35% of cases (7/20), tokens that were not the top-1 match in the original corpus rose to become the best match after evolution. For example, when the original top-5 contained multiple candidate tokens ("building", "facade", "mansion"), the evolutionary search sometimes found that a different token with better surrounding context outperformed the original top-1. This suggests that the fixed corpus's limitation of 20 sentences per token may cause some tokens to be underrepresented due to suboptimal context, even when they would be semantically fitting.

Here, we illustrate three levels of descriptions and their richness obtained from: 1) dynamically generated context, 2) sentences from a fixed corpus, and 3) same token but with lowest scoring context:

| Original (VG corpus) | Evolved (dynamic) | Δsim |
|---|---|---|
| shiny red apples with bright green **leaves** | colorful stamen against green **leaves** | +0.052 |
| a white and black peaked **building** with a peaked roof | grand arched beige **building** | +0.047 |
| the sign reads dried **beef** king | the placard shows assorted **meats** | +0.039 |
| a forest of **lush** trees | bright canopy of **treetops** | +0.031 |
| a nice clear blue **sky** | glistening blue **sky** | +0.023 |

*Table 5.* Examples of dynamic phrase generation improving LatentLens interpretations. The target token is shown in **bold**. Evolved phrases achieve higher cosine similarity to the visual embedding by finding more appropriate contextual framing. Note that evolution can also discover better target tokens (e.g., "beef" → "meats").

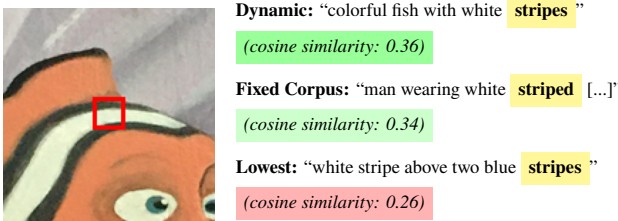

**Dynamic:** "colorful fish with white stripes "
*(cosine similarity: 0.36)*

**Fixed Corpus:** "man wearing white striped [...]"
*(cosine similarity: 0.34)*

**Lowest:** "white stripe above two blue stripes "
*(cosine similarity: 0.26)*

# M. Captioning Quality Evaluation

We evaluate caption quality using DCScore (Ye et al., 2025), a GPT-4o-based LLM judge that rates generated captions on a 1–10 scale across 300 validation images from PixMo-Cap.

**Evaluation rubric.** DCScore evaluates each caption on four fine-grained criteria, each scored 1–10:

- **Faithfulness**: How accurately the caption reflects the actual image content

- **Detail accuracy**: Coverage and correctness of salient visual details

- **Hallucinations**: Absence of non-existent content (higher = fewer hallucinations)

- **Completeness**: Whether the caption captures all key aspects of the image

The overall score is a holistic quality rating that considers all criteria. We provide the judge with both the full image and the generated caption, asking it to return a structured JSON response with all sub-scores.

**Full results.** Table 6 shows the complete caption quality scores for all 9 trained model combinations plus the off-the-shelf Qwen2-VL-7B-Instruct as an upper bound reference.

*Table 6.* Caption quality scores (DCScore, 1–10 scale) on 300 PixMo-Cap validation images. Higher is better. Our trained models average 6.0, while the fully instruction-tuned Qwen2-VL achieves 8.5.

| LLM | Vision Encoder | Score |
|---|---|---|
| OLMo-7B | CLIP ViT-L/14 | 6.60 |
| OLMo-7B | SigLIP | 6.75 |
| OLMo-7B | DINOv2-Large | 4.30 |
| LLaMA3-8B | CLIP ViT-L/14 | 6.79 |
| LLaMA3-8B | SigLIP | 6.95 |
| LLaMA3-8B | DINOv2-Large | 4.46 |
| Qwen2-7B | CLIP ViT-L/14 | 7.08 |
| Qwen2-7B | SigLIP | 6.77 |
| Qwen2-7B | DINOv2-Large | 4.54 |
| *Upper bound:* Qwen2-VL-7B-Instruct | | **8.50** |

**Key observations.** **(1)** DINOv2 models consistently underperform on captioning (4.3–4.5) compared to CLIP and SigLIP models (6.6–7.1), likely due to DINOv2's lack of language supervision during pretraining. **(2)** Qwen2-7B achieves the highest average score (6.13) across vision encoders among our trained models. **(3)** The gap between our models (avg 6.0) and Qwen2-VL (8.5) reflects that our models were trained only on captioning with a frozen LLM, while Qwen2-VL underwent full multimodal instruction tuning. **(4)** Notably, captioning performance does not correlate with visual token interpretability—DINOv2 models show comparable interpretability (Figure 3) despite lower captioning scores.

**Sample captions.** Figure 25 shows example captions from three representative models on the same validation image, illustrating how different model combinations produce captions of varying quality and detail.

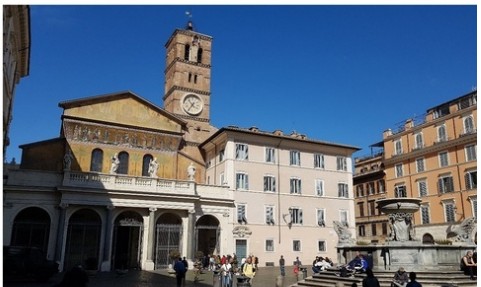

**OLMo + CLIP-ViT** (8/10)

The image depicts a bustling square, likely in Italy, characterized by a mix of old and new architecture. Dominating the scene is a grand, ancient building with a distinctive yellow facade, adorned with a clock tower and a prominent cross. The building features a series of arches and a grand entrance, flanked by two large columns. The square is bustling with people, some seated on benches, others strolling around.

**Qwen2 + SigLIP** (8/10)

The image depicts a vibrant and bustling square in Italy, characterized by a mix of historical and modern architecture. The central focus is a large, ornate fountain, which is the focal point of the square. To the left of the fountain, there is a building with a distinctive clock tower adorned with statues. The sky above is a clear blue, enhancing the overall beauty of the scene.

**LLaMA3 + DINOv2** (7/10)

The image captures a bustling square in a European city, likely in Italy or Spain, with a vibrant mix of architectural styles. Dominating the scene is a large, ornate building with a golden facade and a prominent clock tower. Adjacent to the clock tower is a smaller building with a similar golden hue. In the foreground, a large fountain draws a crowd of people.

*Figure 25.* **Sample captions for the same validation image.** Per-image DCScores shown in parentheses.

## N. Qualitative examples

To demonstrate that our findings are not cherry-picked, we present 20 randomly sampled examples across 5 layers (0, 8, 16, 24, and final layer). For each example, we randomly select one of 10 models (9 trained models plus Qwen2-VL) and show top-3 predictions from each method: **EmbeddingLens** (nearest neighbors in input embedding matrix), **LogitLens** (LM head predictions), and **LatentLens** (ours, contextual nearest neighbors with phrase context).

For LatentLens, we show the top-3 Visual Genome phrases containing the matched token (highlighted in **yellow**), demonstrating the semantic richness of phrase-level interpretations. For baselines, we show the top-3 tokens with EmbeddingLens and LogitLens colored backgrounds. Across all randomly sampled examples, LatentLens consistently provides more interpretable results—the contextual phrases typically describe the visual content more accurately than isolated tokens from the baselines.

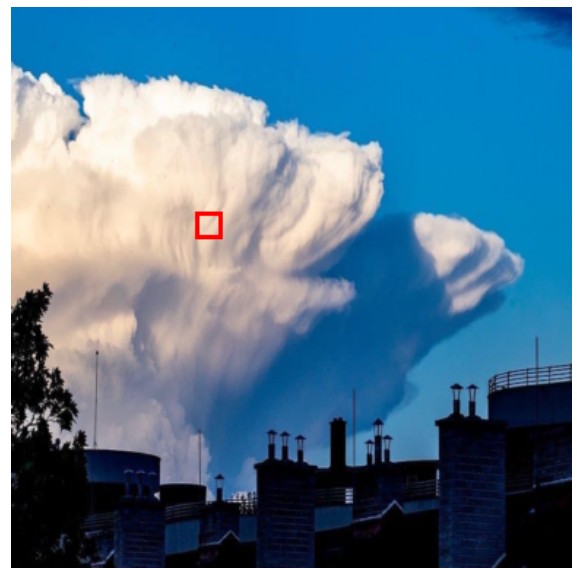

**OLMo+SigLIP**, L0

**LatentLens:**
(1) "bear's paws covered in swirling grass"
(2) "blue tail of plane with white swirls "
(3) "the cat's tail is frizzy ."

**EmbeddingLens:** hairy   Skinny   flashy

**LogitLens:** cir   faucet   rawn

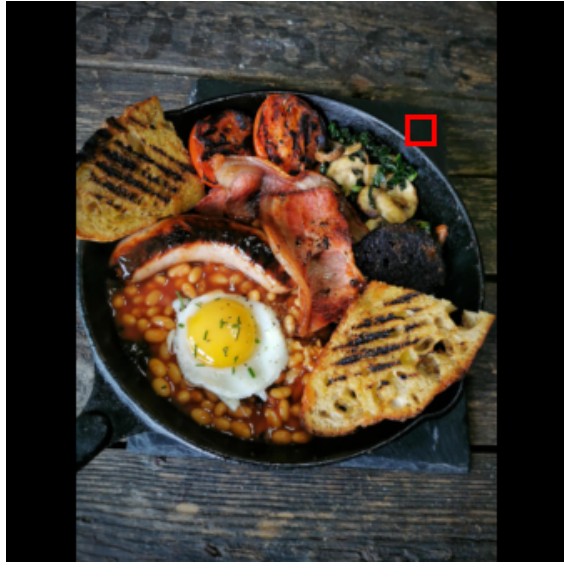

**LLaMA3+CLIP**, L0

**LatentLens:**
(1) "a vase sitting on a square table "
(2) "water glass on a white tablecloth "
(3) "cup and saucer on a wooden table "

**EmbeddingLens:** planted   opies   avou

**LogitLens:** [?]   uzey   tôi

**OLMo+SigLIP**, L0

**LatentLens:**
(1) "train with the letters s. w.n . in white"
(2) "a sign reading "carrer d'en falconer ""
(3) "train with the letters s.w .n. in white"

**EmbeddingLens:** Vanessa   [?]   Wolf

**LogitLens:** hausen   ens   experiment..

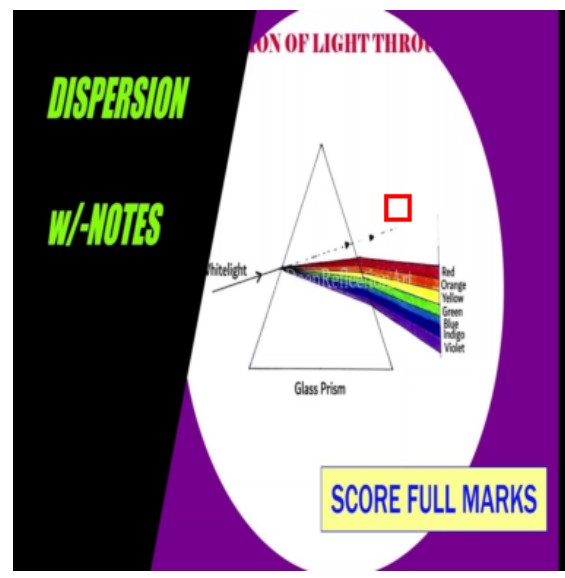

**OLMo+SigLIP**, L0

**LatentLens:**
(1) "the water had the sun gleaming down"
(2) "...ncluding a dissassembled bicycle and..."
(3) "bathing suit for swimming and suntanning ."

**EmbeddingLens:** [HI]   218   Education

**LogitLens:** erne   rone   crest

*Figure 26.* **Layer 0 (input):** Four randomly sampled visual tokens at layer 0.

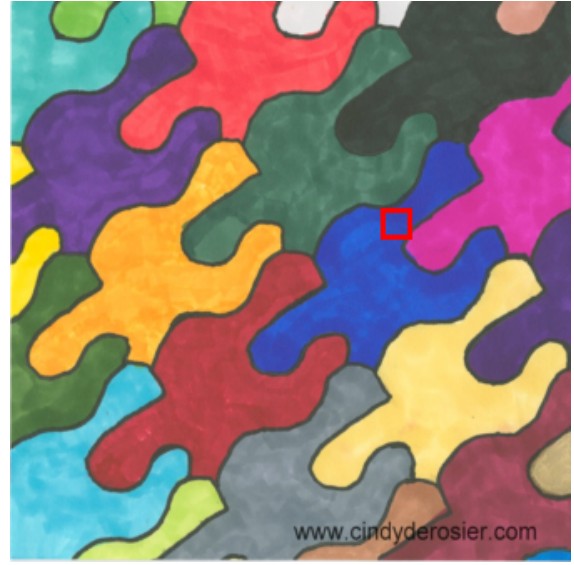

**Qwen2+DINOv2**, L8

**LatentLens:**

(1) "...s, probably 3 [ **though** the one beneat..."

(2) "a second adult **bows** low in to inspect t..."

(3) "...e bear+person [ **though** that might not..."

**EmbeddingLens:** -c [?] (translate..

**LogitLens:** tô Según ==========..

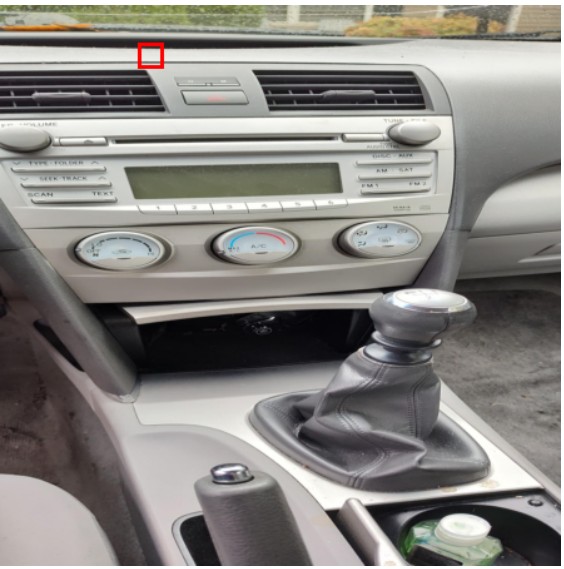

**Qwen2-VL**, L8

**LatentLens:**

(1) "controls are on the center **console** "

(2) "dials on motorcycle **dashboard** glow red"

(3) "controls are on the center **console** "

**EmbeddingLens:** [TH] .urlencoded .DropDownL..

**LogitLens:** [?] <nav Jihad

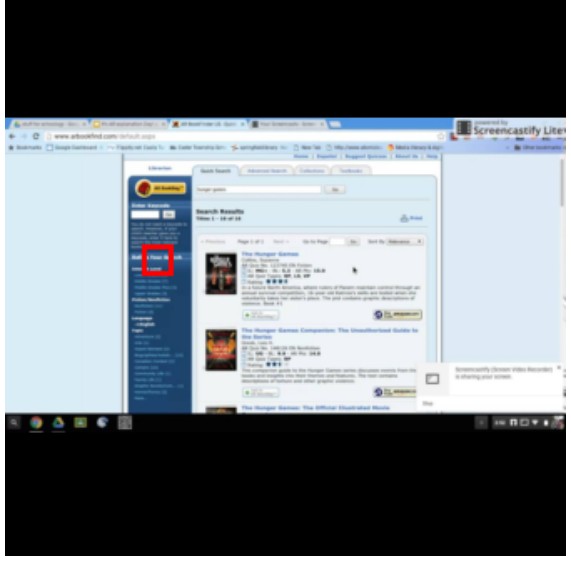
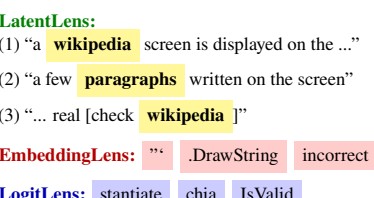

**Qwen2+CLIP**, L8

**LatentLens:**

(1) "a **wikipedia** screen is displayed on the ..."

(2) "a few **paragraphs** written on the screen"

(3) "... real [check **wikipedia** ]"

**EmbeddingLens:** "' .DrawString incorrect

**LogitLens:** stantiate chia IsValid

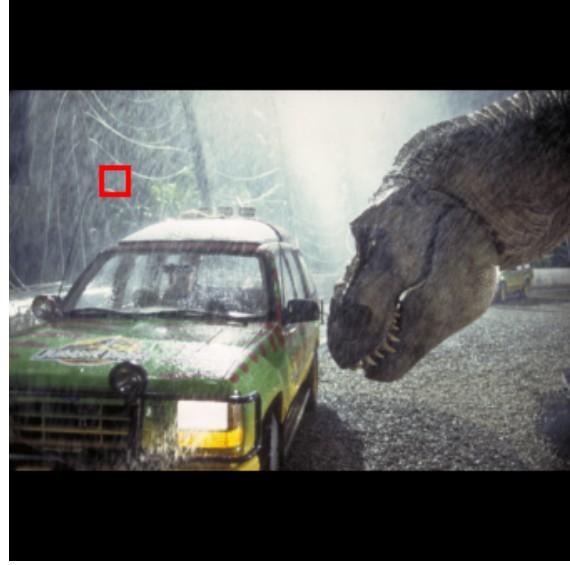
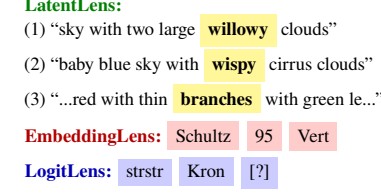

**LLaMA3+CLIP**, L8

**LatentLens:**

(1) "sky with two large **willowy** clouds"

(2) "baby blue sky with **wispy** cirrus clouds"

(3) "...red with thin **branches** with green le..."

**EmbeddingLens:** Schultz 95 Vert

**LogitLens:** strstr Kron [?]

*Figure 27.* **Layer 8 (early-mid):** Four randomly sampled visual tokens at layer 8.

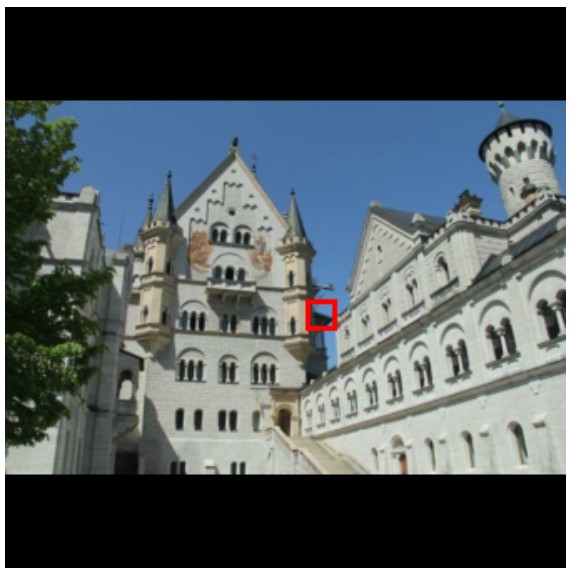

**Qwen2+CLIP**, L16

**LatentLens:**
(1) "stone chimney attached to a **house** "
(2) "pole and trees in front of **building** "
(3) "low, grey roof of **residence** ."

**EmbeddingLens:** (translate.. ([... .isRequired

**LogitLens:** chantment [?] onUpdate

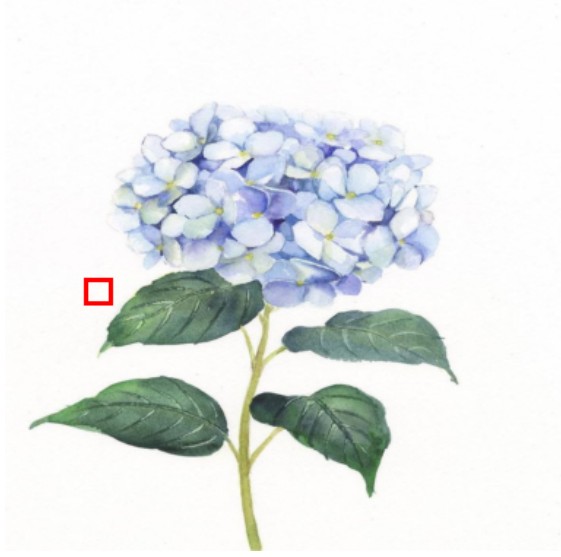

**Qwen2+SigLIP**, L16

**LatentLens:**
(1) "framed pencil **sketch** on wall"
(2) "pencil **sketched** artwork of a fruit basket"
(3) "framed **drawing** to left of the bed"

**EmbeddingLens:** There div -cut

**LogitLens:** [?] [?] fdc

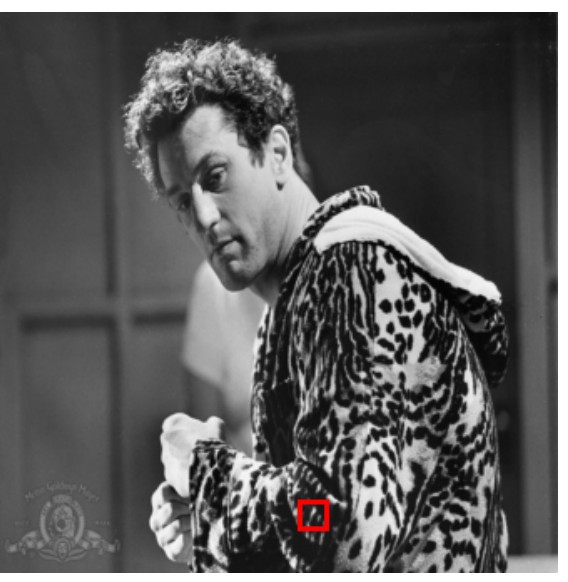

**Qwen2+DINOv2**, L16

**LatentLens:**
(1) "...ith black polka **dots** "
(2) "...er with a blue **stipe** "
(3) "man wearing a white shirt with blue **stipe** "

**EmbeddingLens:** .COMP Specifies rà

**LogitLens:** CascadeType =nil oppon

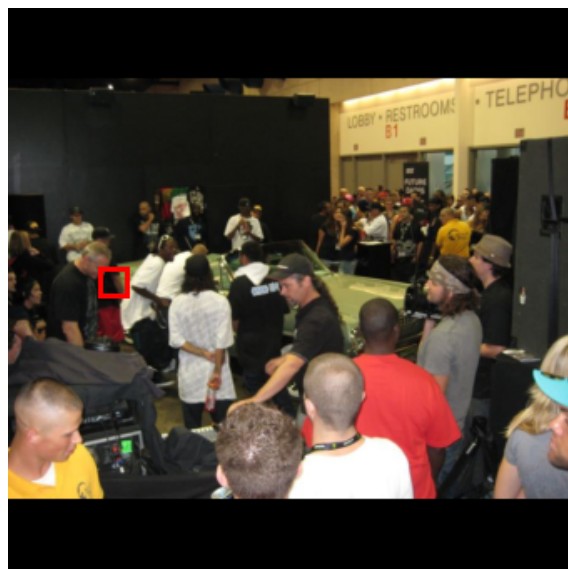

**OLMo+CLIP**, L16

**LatentLens:**
(1) "man with balding head **standing** near cart"
(2) "...diners, four of **whom** are toasting, a..."
(3) "older man **sitting** on wooden bench"

**EmbeddingLens:** him men Boys

**LogitLens:** minority mostly oya

*Figure 28.* **Layer 16 (middle):** Four randomly sampled visual tokens at layer 16.

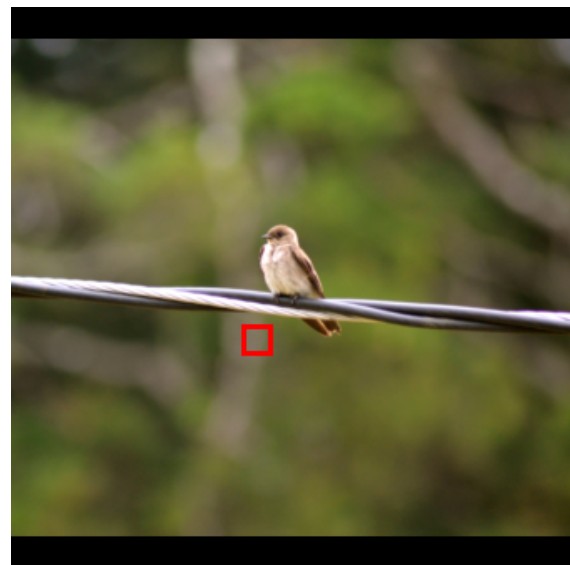

**LLaMA3+CLIP**, L24

**LatentLens:**
(1) "...leaves against `muted` background"
(2) "...hink black and `white` stripes"
(3) "...st a wall with `muted` white lights ."

**EmbeddingLens:** white | White | allied

**LogitLens:** .scalablyt.. | UnitOfWork | [?]

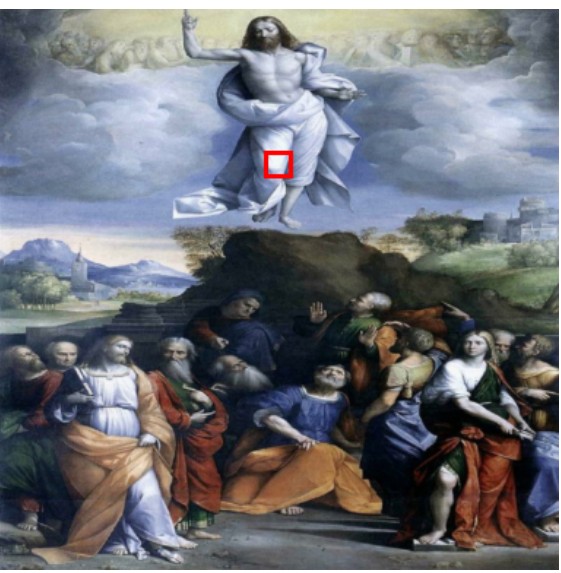

**Qwen2+SigLIP**, L24

**LatentLens:**
(1) "man wearing `white` baseball cap."
(2) "lady in `off-white` sweater"
(3) "back of woman wearing a white `turtleneck` "

**EmbeddingLens:** .leading | machining | GetX

**LogitLens:** _unregister | [?] | ("/

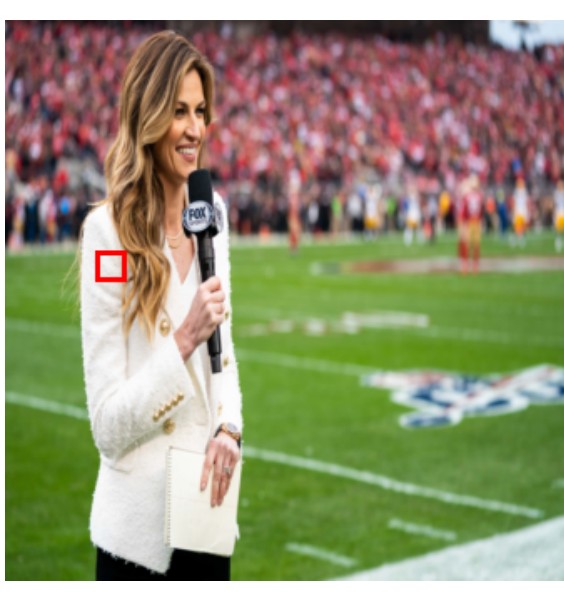

**LLaMA3+DINOv2**, L24

**LatentLens:**
(1) "a woman in a tan `blazer` "
(2) "man wearing grey `coat` "
(3) "woman wears an `off-white` ski jacket"

**EmbeddingLens:** o | k | Drawable

**LogitLens:** classpath | [?] | [?]

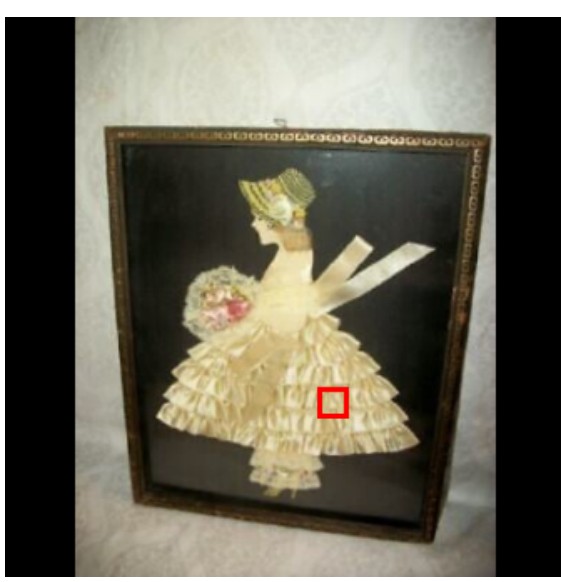

**LLaMA3+CLIP**, L24

**LatentLens:**
(1) "... lime green , `apricot` orange , turqu..."
(2) "blue, `bergundy` and gray coat."
(3) "tan/ `yellow` dog laying across young girls lap"

**EmbeddingLens:** Gold | gold | white

**LogitLens:** GOLD | ConverterF.. | HEST

*Figure 29.* **Layer 24 (late):** Four randomly sampled visual tokens at layer 24.

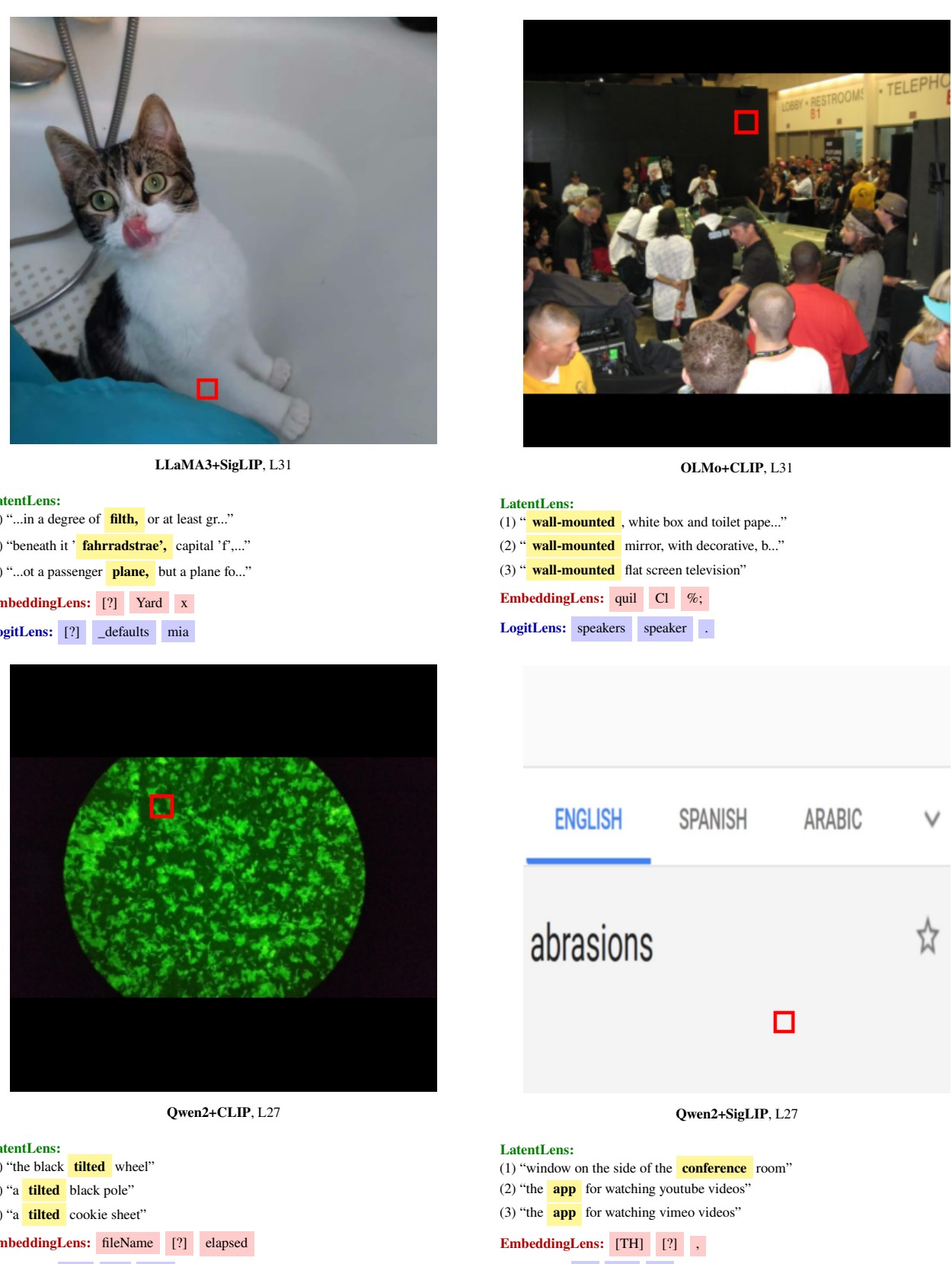

**LLaMA3+SigLIP**, L31

**LatentLens:**
(1) "...in a degree of **filth,** or at least gr..."
(2) "beneath it ' **fahrradstrae',** capital 'f',..."
(3) "...ot a passenger **plane,** but a plane fo..."

**EmbeddingLens:** [?] Yard x

**LogitLens:** [?] _defaults mia

**OLMo+CLIP**, L31

**LatentLens:**
(1) " **wall-mounted** , white box and toilet pape..."
(2) " **wall-mounted** mirror, with decorative, b..."
(3) " **wall-mounted** flat screen television"

**EmbeddingLens:** quil Cl %;

**LogitLens:** speakers speaker .

**Qwen2+CLIP**, L27

**LatentLens:**
(1) "the black **tilted** wheel"
(2) "a **tilted** black pole"
(3) "a **tilted** cookie sheet"

**EmbeddingLens:** fileName [?] elapsed

**LogitLens:** -left left right

**Qwen2+SigLIP**, L27

**LatentLens:**
(1) "window on the side of the **conference** room"
(2) "the **app** for watching youtube videos"
(3) "the **app** for watching vimeo videos"

**EmbeddingLens:** [TH] [?] ,

**LogitLens:** [?] <?= [?]

*Figure 30.* **Final layer:** Four randomly sampled visual tokens at the final layer (31 for OLMo/LLaMA, 27 for Qwen2).

# O. Behind The Scenes

The goal of this section is to make science more transparent and engaging, showing not just the polished paper at the end but also all the detours and lessons learned.

## O.1. From start to finish

**The pivot.**   This project started last winter (December 2024) when the first author, originally working on vision-and-language models in general, was determined to pivot towards *understanding* models and fundamental science. So interpretability was the natural direction to look into, from afar it had been intriguing to follow the field the past few years. But just doing interpretability for the sake of interpretability did not feel like the right approach. So what was an actual fundamental question that many people would care about, ourselves included, where interpretability could naturally help?

**Brainstorming.**   The first author tried to present too many potential ideas at once in their lab meeting. There was not enough time to present them all, and it would not have been a fun presentation to follow with too many disjoint pitches. So a lab colleague simply asked: "Which one of these questions are you genuinely excited about?" And this is how we ended up with this paper and the research question we ask: How can frozen LLMs possibly make sense of visual tokens? Do they look like language tokens to the model? We started with this simple question but the first author could not have possibly imagined all these new insights we would stumble upon, such is the process of science. It was really a situation of unknown unknowns: the kind of experiments and ideas we would eventually end up with were inconceivable at the time, and how with each new experiment, three new options opened up. As a result writing the paper in 8 pages was very challenging and three sections had to be cut even from the appendix.

**Don't trust assumptions.**   If the first author had to take away just one lesson from the project, it would be to never trust long-held assumptions (either by oneself or the field). We had assumed that visual tokens were not really interpretable with nearest neighbors from the embedding matrix, and several papers hinted in that direction. In retrospect, we should have simply tested this empirically across several models from day one. Instead, the first author read papers about anisotropy and other interpretability literature, prematurely concluding that embeddings from different modalities live in entirely different narrow cones.[13] So, then the question became: If these tokens are not interpretable and live in different subspaces, maybe we have to perform more linear algebra and embedding space tricks to show how they relate to text? Rotate them around, learn some simple mappings, specific subspaces, ...?

**The Mosaic Dataset.**   Eventually, the first author hypothesized: Maybe visual tokens are not directly mapped to interpretable words (measured by NNs from the embedding matrix) because real data is messy? Visual objects in a scene span many tokens, or a single token might represent several objects and different attributes. So it would be no surprise that they don't just cleanly map to a single concept. But what if we could simplify this situation? What if we could make the data we train on simpler and simpler until we see very predictable nearest neighbors. This is where the project went "off-track" for a while and we started experimenting with toy datasets we coined *Mosaic*, where the model gets as input an image like this:

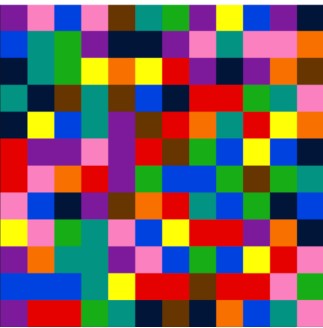

The LLM would then be trained to predict the sequence of color words one after the other ("red blue red pink ..."), essentially "copying" each visual token's corresponding color to the text space. We observed some interesting quirks, but surprisingly

---

[13]We even ran lots of experiments on anisotropy and concluded that effects reported in other papers seemed overly simplified. At this point I wouldn't feel confident saying that different modalities live in different narrow cones inside e.g. an LLM.

not many interpretable nearest neighbors (i.e. color words from the embedding matrix). In fact, fewer than on the natural images we report in this paper. We even ran various causal patching experiments, studied high-norm outliers, etc! They all seemed interesting at the time, but again, simply testing our assumptions early would have saved us these detours. At this point we had already adopted the Molmo codebase for training models, since we cared about how *frozen* LLMs make sense of visual tokens. But most VLMs you can take off-the-shelf unfreeze the LLM weights at some stage of training, which is why we opted for our controlled training setup that the final paper still builds upon.

**Hope.** The first author does not remember when exactly we finally questioned our assumptions, and empirically tested the interpretability on natural images. But sometime around July, we started exploring interactive demos of natural images with the top NN from the embedding matrix. Since we happened to explore OLMo+CLIP-ViT, we were surprised to see so many meaningful words. The cosine similarity was low (between e.g. $0.07$ and $0.15$), much lower than what LatentLens eventually yields. Nonetheless this was encouraging, but it soon became clear that automating the judgment of whether an NN is interpretable is not trivial.[14]

**A first glimpse of the final story.** With these exciting findings (40% to 60% of tokens in OLMo+CLIP-ViT are interpretable), the initial story of the paper was roughly: Visual tokens at layer 0 are more interpretable than some would assume! In the background we kept working on directions that did not end up in the main paper or even appendix: 1) We conducted ablations under which conditions this interpretability would increase or decrease, now in Appendix D, though without any trace of these initial EmbeddingLens results. 2) We always wondered what these strange non-interpretable tokens encode... They often had the same EmbeddingLens nearest neighbors, they were often in non-salient background regions. Were they something akin to task vectors or register tokens? At the time this seemed like an interesting enough story to publish: Past work assumes visual tokens are rarely interpretable at the input via EmbeddingLens but here we show for some models that is not the case. We show ablations, we show patching experiments. Some qualitative results. It is good we kept exploring further: Some co-authors kept wondering, especially SR and MM: What happens to visual tokens at LLM layers after the input? And that's where the final paper we now have started taking shape. We adopted EmbeddingLens and LogitLens not just at the beginning or end of the model but throughout. And eventually we wondered: What if we use contextual embeddings instead of static embedding matrices? (Thank you MM and Elinor Poole-Dayan for the nudge!)

## O.2. Lessons and Reflections

**Automation.** This project co-occurred with the rise of Cursor and Claude Code. Especially interactive demos for quick exploration are now much easier to build, crucial for projects like this one. Due to the speed of experimentation, a lot of ideas on the side did not make it into the paper but can now be easily continued as follow-up work.

**Lessons on science.** As mentioned above, testing the simplest assumptions early on is something the first author will keep as a guiding principle for future research. The field knows less than one would expect. Models change constantly, experiments might look different with slightly different setups. Re-run what others have seemingly done before.

**Interpretability is fun.** The process of open-ended discovery is very enjoyable and naturally leads to interesting brainstorming sessions with colleagues and friends. The first author highly recommends working at least on one interpretability project in one's research journey. The community is quite unique, reflective and open to good ideas (ideally) regardless of whether they are immediately useful downstream. The first author recently wrote a blog post, which goes into detail about how the pivot to interpretability last year went: Better late than never: Getting into interpretability in 2025.

**Personal.** This project carries a lot of meaning. It will be the final chapter of the first author's PhD and was in many ways a perfect ending to this five-year long journey. I have rarely felt so proud of a work, and again not just because of the final product, but the whole process: Every single collaborator on here was fantastic, either as my closest friends throughout the PhD or as recent cherished connections and mentors. A special shout out goes to MM and DE who both put a lot of care into this paper. This project also perfectly encapsulates the first author's strengthened conviction to stay in academia for the long haul. In a way it represented what academia ideally could be and what we should strive for it to be. Looking back at other projects, never before did I have so many enjoyable interactions when sharing what we are working on. Even before the work is now getting out, I gave four talks about it (three in-person), and many more Mila coffee table chats. It made me

---

[14]It would take another 3 months to develop the full LLM judge we eventually relied upon.

realize how science is so much more than papers, and papers are just one way to let others know what you found. One can give talks, write blog posts, make videos, tell others about your work, write it into the snow, showcase it as a demo.

## O.3. Post-rebuttal

The paper was accepted at ICML 2026 and in the spirit of this section we will also add details about the reviewing process and what we updated afterwards. The full reviews and our responses can be found here on OpenReview. The scores improved from 3344 to 3445 (out of scale of 6) during the rebuttal period.

**Highlights and main critiques of the paper.** The most interesting comments came from reviewer u6Pj about faithfulness of our LatentLens interpretations: How do we know if the top-5 results LatentLens gives us are actually influencing model behavior and not just some side-effect? The short answer is: we don't. The longer answer is what we wrote in the rebuttal. Warning: It is several paragraphs and is influenced by my own eagerness to convince a reviewer of the merit of our findings. Feel free to skim.

> *First, we would like to thank for the reviewer for genuinely engaging with our work and the broader field of interpretability. We as authors have had similar debates over the last year around "What is the point of interpretability/analysis work?" and how can the field improve.*
> *While it seems the reviewer has mostly made up their mind, which we have to accept, that faithfulness is not just an option, we would like to share briefly our (subjective) perspective and make some case that the paper is a solid scientific contribution:*
> *When the project was ongoing, several months before submission, we/I were also at this crossroad: Should we focus on downstream performance/outputs of the model, or purely on the main claim/RQ of the paper? We did decide to run downstream experiments primarily testing whether the interpretable visual tokens (using cosine similarity as a proxy) are more important for downstream captioning than the non-interpretable ones. Concretely, we would zero-out e.g. either 30% of interpretable or non-interpretable tokens. However, some LLMs behaved quite strangely and results were too preliminary to share. But more broadly, when the first author reflects on existing analysis-flavored papers, it can seem worse to "squeeze in" some last section in the paper with often very early results but presenting them as "our paper shows that this will improve models/safety". This might have arguably done more harm than good for fields like interpretability (now faced with very high expectations to fix and control models), when authors feel every paper has to claim improvements/safety downstream, instead of simply showing evidence for the identified phenomena. We would rather prefer that some papers (like ours) dedicate themselves to deeply investigating some regularity or pattern in the model, and crucially also ensuring that the results generalize across many models, and many ablations. Then another paper can dedicate not just 1 page but a whole paper to studying downstream effects. As long as the first paper does not claim much about downstream performance, we think this is a fine way for science to advance. For example, hallucination detection is a complicated enough problem that it is hard to focus on in just a small part of the paper, and it should not be done quickly at the end of the project. We think there are many applications of LatentLens that could be more about broad insight than fixing a downstream problem (but hopefully also someone comes and shows if it's useful for e.g. hallucination detection!): Someone studying latent thinking tokens might want to know if these tokens are still close to discrete language throughout the model; or someone studying multi-linguality might want to know at which layers and on which inputs LLMs represent non-English tokens closest to English latent representations (and which ones those are).*
> *Finally, in the words of reviewer BNgn: We genuinely think this finding could be a "Big, If True" finding. Thus, we wanted to make sure that it is in fact true and laser-focus on that alone: with controlled training setup, ablations, manual inspection of many examples, demo, etc. Even then, reviewers asked for more models and other empirical justifications, showing us that we can do better even in this regard (e.g. now having 3 off-the-shelf models and not just 1, see BNgn response). Often, interpretability papers can fall short here instead of faithfulness: using only 1-2 off-the-shelf models, not ablating the training (understandably with the costs).*
> *We understand that the reviewer has this strong and justified focus on faithfulness (the field has had many issues with it in the past), and there might just be a fundamental divide in opinions here. We hope sharing our perspective was at least interesting and maybe helpful.*

I am quite optimistic that there will be causal connections to outputs, e.g.: If we ablate certain visual tokens to push them closer to other nearest neighbors instead, the resulting caption or QA answer should often change. But someone still has to run these experiments rigorously.

What were the other main critiques of this work?

1. How sensitive are our results to the corpus choice (captions from Visual Genome) and the size? This is why the paper now contains Section 4.5 and why we also released a more general pool of contextual embeddings (beyond just visual concepts) in our GitHub repository.

2. The memory and time cost to use LatentLens: It is a bit more effort than running LogitLens or EmbeddingLens. But

we do think it is worth it to get these better interpretations, and especially with our results in Table 1 as well using a more targeted corpus, the memory footprint is not massive. In our naive implementations with around 3M captions, we ended up with 26G per model.

3. More models! We agreed with reviewers that just showing one off-the-shelf model (Qwen2-VL-Instruct 7B) was not enough (although we also have 9 controlled trained models). So we added 5 more models, resulting in our final version of Section 4.4.

**The future of the paper.** The paper will be presented in Seoul, Korea, and has been widely shared during many talks in the last months. As of now, we are actively pursuing several lines of follow-up works to explore the potential of LatentLens: other types of non-language tokens (e.g. audio or latent thinking), video instead of single-image input, etc. And yes, also the faithfulness question will and should be explored, either by us or others!

