# OpenReview forum: "LatentLens: Revealing Highly Interpretable Visual Tokens in LLMs"
_ICML.cc/2026/Conference — ICML 2026 regular_

### Official Review · Reviewer_BNgn · 2026-02-23

**Soundness:** 2
**Presentation:** 3
**Significance:** 4
**Originality:** 3
**Overall Recommendation:** 4
**Confidence:** 4

**Summary:**

This paper introduces LatentLens, a new method for interpreting the visual tokens in Vision Language Models. A large corpus of text captions is first encoded using the language model. To interpret a visual token, its hidden representation is compared against the encoded corpus to retrieve the top-k most similar text tokens, which ideally capture the visual token's semantic meaning. The authors train VLM projectors for different combinations of language models and vision encoders, and demonstrate that LatentLens is more interpretable than two prior methods, LogitLens and EmbeddingLens, according to a VLM judge.

**Compliance With Llm Reviewing Policy:**

Affirmed.

**Final Justification:**

The rebuttal has addressed concerns regarding generalizability to off-the-shelf models, compute costs, and clarified a typo. However, it also surfaced new concerns regarding the fairness of the VLM judge-based metric. The authors acknowledge this is a concern, but do not take or propose concrete steps to fix it. Overall, I'm keeping my initial positive score (4).

**Key Questions For Authors:**

Please see the weaknesses.

**Limitations:**

The authors did not discuss limitations.

Technical Limitations include:
1. The need to embed a large and representative corpus of text captions for each VLM.
2. The lack of experiments related to image resolution.
3. Only testing one off-the-shelf VLM.
4. The use of a VLM judge, which is only moderately correlated with the authors' judgment (Cohen's Kappa 0.68).
5. Lack of independent human annotators.
6. Potential bias of VLM judge in favor of LatentLens' full-word text vs other lenses' token text.


I do not foresee any negative societal impact.

**Strengths And Weaknesses:**

I would say this is certainly a "Big, If True" paper. LatentLens appears to be a solid improvement over LogitLens, which has been widely used in the VLM interpretability community. Given the potential impact, the roster of experiments seems to be somewhat limited.


Strengths
1. LatentLens is substantially more interpretable than LogitLens and EmbeddingLens.
2. LatentLens can generate sentence-level descriptions of visual tokens, while prior methods only generated tokens that often lack context or may even be incomplete.


Weaknesses
1. The authors do not test the effect of image resolution, different image-patching methods (such as Gemma3's pan-and-scan, Qwen's Pixel Shuffle, and DeepStack) on LogitLens. The main results are from VLMs, where only the projection layer has been trained. When training state-of-the-art VLMs, it is common to train the language model as well. It is unclear if LogitLens would work on such VLMs since the authors only tested Qwen2-VL-7B.

2. The authors do not inspect LogitLens' ability to recover high-level semantics (for example, given the picture of a car, the tokens could map to "wheels", "door", etc, instead of "car" ).

3. Section 3.2 mentions that the visual token h_l at layer l is compared to all corpus tokens r_l, also from layer l. However, Figure 4 shows that visual representations from the early layers do not align with caption text representations from the early layers. Given this mismatch, it is unclear to me why even early layers are judged as highly interpretable in Figure 3c and Figure 4.

4. Since LogitLens and EmbeddingLens recover only tokens that are sometimes word fragments instead of complete words, it is possible that the VLM judge is underestimating their interpretability. For example, given a picture of a soda can, LogitLens might recover "alum"(inum can), which a VLM judge might dismiss as non-interpretable. Since the authors provide full-word descriptions for LatentLens, this might bias the VLM judge in favor of LatentLens.

---

> ### Author Rebuttal · Authors · 2026-03-31
>
> We thank Reviewer BNgn for their positive comments and raising valuable concerns. We especially appreciate that they found our paper to be a “Big, If True” paper; we had the same reaction. We will now briefly address main concerns and questions:
>
> ## VLM Coverage
> Our main focus is a controlled and simplified setting to disentangle different factors easily. Most off-the-shelf VLMs differ in more than one way (preprocessing, backbones, data, unfreezing, …). In this controlled setting we study 3 vision encoders and 3 LLMs, yielding nine different model combinations.
> However to show that LatentLens works across any VLMs, we ran it on two additional off-the-shelf models (so a total of three with the Qwen2-VL already in the paper) and find similar trends of LatentLens working well across all three.
>
> Interpretability judgement results avg. across layers:
>
> | Model | EmbeddingLens | LogitLens | LatentLens |
> |-------|--------------|-----------|------------|
> | LLaVA-1.5-7B | 40.8% | 34.9% | **55.2%** |
> | Molmo-7B* | 44.1% | 35.7% | **85.9%** |
>
> *We only interpret the full-image crop, not the additional high-res crops, for fair comparison.
>
> Interestingly, a) our observed mid-layer leap phenomenon (where early/input visual tokens get contextual NNs from later LLM layers) also holds on the 3 off-the-shelf models; b) we find slightly more variance across layers than in the original paper. This is why the overall LLaVA-1.5-7B results are lower with early layers around 45% and mid-layers at 70%.
>
> ## Different design choices, e.g. image resolution
> VLMs have so many different design choices (e.g. also the type of adapter), that we could not choose all and would not have been able to disentangle factors. We aimed for the most common design choices (such as MLP mapping used across e.g. Molmo, Qwen-VL families, etc.) and for simplicity. However we will discuss resolution in the camera-ready, since it might also play a role in why we see higher interpretability for Molmo-7B above (the full-image crop is only 12x12).
> We are optimistic that LatentLens will generally work well across design choices, with small but interesting variance that future work can explore.
>
> ## High-level semantics
> The reviewer notes: “No high-level semantics inspection [...]".
> We encourage the reviewer to briefly explore examples in Appendix J, or "G. Fine-grained Interpretation Analysis" where we explore such details. Overall, our judge is explicitly designed to consider high-level semantics. We even allow the judge to identify “abstract” interpretations such as “Italy” or “Renaissance” for a mediterranean-looking town square. Just by manually inspecting ten samples of LatentLens top-5 results, the majority contained diverse high-level semantics such as “fitness”/”workout” for a man in the gym or “reading”/”writing” on a screenshot with text in it.
>
> ## Misleading indexing of layers in method description
> We thank the reviewer for catching this. We had already corrected the indexing of the layer in the variables we defined shortly after submission and will carry this correction forward to camera-ready. As you rightly noted, we allow a visual token to get its contextual NN from \*any\* LLM layer. This is crucial to our method and to our insights and allows us to e.g. conclude “visual tokens at layer 0 are closest to later-layer LLM representations and the residual stream may allow them to be carried forward without too much change to those later layers”.
>
> ## VLM judge bias toward LatentLens
> We agree that there can be biases in LLM judges. But in this case we would argue that this is actually simply a strength of our method: It is designed to provide full-word or often even full-sentence interpretations. If LogitLens and EmbeddingLens cannot do this, it makes interpretability for humans and automatic metrics harder. In cases where the full-word can obviously be derived from the sub-word such as “alum” from “aluminium”, the LLM judge would usually recognize this and in its reasoning trace make a good guess (“the shown interpretation likely refers to X”).
>
> ## Need to embed a large, representative corpus for each VLM
> We agree that this can be an initial burden but note that it is a one-time effort at the beginning (several hours) and since other reviewers had similar comments, we ran an ablation showing that randomly sampling a small subset of contextual embeddings leads to very similar results (e.g. only starting to drop around sampling at 1%), see response to tiFA for detailed results.
> We also release a smaller more curated corpus, also with pre-computed embeddings for popular models (again see tiFA response).
>
> ───
>
> We hope the additional tested VLMs and clarifications addressed your main concerns. If so, we would appreciate this being reflected in your rating.

---

> > ### Author Rebuttal · Reviewer_BNgn · 2026-04-02
> >
> > I have updated this acknowledgement after the authors responded to my follow-up questions. Overall, I have a better understanding of the paper, but I do not fully agree with the definition of *interpretable* used by the authors. The authors consider mislocalized and hallucinated descriptions as *interpretable* in the sense that they provide some insight into the VLM's inner workings. This is a somewhat loose definition of interpretable, buried in the paper.
> >
> > Due to this evaluation limitation, I am keeping my score.
> >
> > ---
> > **Previous Response**
> >
> >
> > Thank you for the response. I have some additional concerns
> >
> > ### 1. VLM-Judge Overestimating LatentLens interpretability
> >
> > Out of the 20 examples in Appendix J, only 8 images had LatentLens results that I think are interpretable, 4 are completely wrong, and the remaining 8 have issues such as mislocalization (white background of a sketch labelled as "sketch") or vagueness (wall being labelled as "wall-mounted"). I thank the authors for showing non-cherry-picked examples, but this has heightened my concerns that VLM-Judge overestimates LatentLens, given its moderate agreement with humans (Cohen's K = 0.68). Instead of a singular agreement metric, could you report if VLM-judges are consistently over-estimating interpretability compared to humans?
> >
> > Also, in Appendix J, which layer are the reported LogitLens/EmbeddingLens tokens from?
> >
> > ### 2. "actually simply a strength of our method" framing
> >
> > I disagree with this take in the context. When using LatentLens/LogitLens for downstream tasks, it's true. But as a diagnostic tool to study the representations, the bias underestimates LogitLens, and the comparisons using VLM-judge do not seem fair. Also, sub-word tokenization is a language-specific problem, and I've found that the middle layers' Logit Lens on Qwen-2-VL work much better than shown in Figure 5, since they often decode to Chinese characters that are not subword tokenized.

---

> > > ### Author Response · Authors · 2026-04-05
> > >
> > > Thank you for further engaging.
> > >
> > > **Regarding the VLM judge:**
> > >
> > > In the paper (3.3) we explain that we "prompt the judge [...] to classify such cases as concrete (directly visible), abstract (conceptually related), or global (present elsewhere in image)". The latter two cover some of  cases you mention from the Appendix J. We do think this is appropriate, and apply the same criteria to our baselines: We do not want to make any claims about whether the model is representing the visual tokens exactly "correctly", i.e. whether the concept is localized or whether it is a very literal/direct concept (e.g. the object name). We simply care if it makes sense to a human and is semantically related to the image or the region of the image. Sometimes we even find meta-concepts such as "blurry" on a blurry image (i.e. *abstract* category). Prior embedding space methods would instead often yield **completely** random NNs, which does not give much useful insights into model representations.
> > >
> > > Quantitvely, we just annoted 300 additional examples (100 for 3 models at layer 16) to enable a direct comparison to the numbers we show in the main Figure 3. Result:
> > >
> > >   | Model | Human (interp. %) | LLM Judge (interp. %) |
> > >   |---|---|---|
> > >   | OLMo-7B + ViT-L/14-336 | 83% | 75% |
> > >   | Qwen2-7B + DINOv2-L-336 | 83% | 80% |
> > >   | LLaMA3-8B + SigLIP | 63% | 63% |
> > >   | **Average** | **76%** | **73%** |
> > >
> > > Under inspection some of the examples where only the human said "interpretable" were general meta-concepts such as "image" or "background" as LatentLens NN. We acknowledge this can become subjective on rare edge cases but argue that the overall trend and insights from the paper still stand.
> > >
> > > **Subwords:**
> > >
> > > We agree that we would underestimate LogitLens if we are not careful. But humans/judge try to give the benefit of the doubt: If we saw "amster" on an image of a Dutch building, we would assume it relates to Amsterdam and say "interpretable". But if we simply saw "am" as a subword, how is one supposed to judge whether this is interpretable, even as a diagnosis for representation spaces as you point out? "am" can occur in many words like "amputate", "america" or "ampersand". Likewise, we auto-translated Chinese characters and also checked with Chinese colleagues if our translations were accurate; our GPT-5 based judge would also take Chinese tokens into account in its reasoning. So in cases where LogitLens shows interpretable Chinese tokens, we would count it.
> > >
> > > ---
> > >
> > > Overall, we are happy to elaborate on these points in the paper and fully agree that these are important questions. If the reviewer thinks this is a significant and interesting VLM interpretability finding, we would appreciate this being reflected in the score.

---

### Official Review · Reviewer_u6Pj · 2026-03-11

**Soundness:** 2
**Presentation:** 4
**Significance:** 2
**Originality:** 3
**Overall Recommendation:** 3
**Confidence:** 4

**Summary:**

This paper proposes a method for interpreting the latent representations of visual tokens in vision-language models, where the LLM backbone and the visual encoder are frozen and only the connector, which maps each image patch to a visual token, is trained. The idea is to first embed a large set of text descriptions to obtain contextual token representations that can potentially interpret a visual token representation. For a given visual token, the top-k similar contextual tokens representations are retrieved with their original texts. The top-k interpretations are evaluated by an independent LLM judge.

**Compliance With Llm Reviewing Policy:**

Affirmed.

**Final Justification:**

As I wrote in the review and in the comments during the rebuttal, I acknowledge the direction and the breadth of the paper. My concern is only the soundness and the practical impact. Without showing the faithfulness of the generated explanations, it is difficult to appreciate the usefulness of the explanations. I'll therefore keep the score.

**Key Questions For Authors:**

- Can you show that the interpretations by LatentLens are faithful? If not, how can we use the interpretations?
- Why aren't stronger baselines used or an attempt made to make the baselines stronger?
- What is the text data, from which the contextual representations are obtained, can only cover a fraction of interpretations? What if the fraction mostly consists only of only plausible interpretations and misses many faithful interpretations?
- In the paper only training-free approaches are used as baselines. Why? Wouldn't approaches that decompose the latent representations into (linear) combinations of interpretable concepts (e.g., by using SAEs) be strong baselines?

**Limitations:**

No. Evaluations are based on external VLM judge which may be biased towards concepts that are directly observable in images. Concepts that are not in the image may be difficult to be evaluated.
The authors do not make causal claims about the interpretations, i.e., the interpretations may be plausible but not faithful to what is really encoded in the latent representation.

**Strengths And Weaknesses:**

The paper tackles the problem of interpreting the latent representations of visual tokens in a VLM that is based on frozen LLM backbone and image encoder. The paper is well presented and easy to follow, a particular reason being that the main idea is intuitive. All in all, the presentation of the paper is very good. However, I have the following concerns that the authors could address in their rebuttal:
- Correctness guarantee: even though a VLM judge accepts an interpretation as correct, there is no guarantee that the judgement was correct and a VLM judge cannot measure the faithfulness of the explanation. Deeper investigation in this direction would be necessary.
- Baselines may not be strong: since LogitLens is based on the "unembedding" layer, it may not very well compatible with intermediate layers. This issue is reflected in the experiments: in Fig 3b, even thought LogitLens leads to high percentage of interpretable tokens comparable to LatentLens for later layers, it does not perform well for earlier layers. This issue was addressed by Tuned Lens [1], which trains a non-linear map instead instead of using the linear unembedding layer. This may resolve the incompatibility issue to some extend. The same idea of using a non-linear map can be applied to the EmbeddingLens approach, e.g., by training a non-linear map on text data such that the input token embedding is the closest to its latent representation. Then for a visual token representation the most similar input token can be found by using the non-linear map. In summary, stronger baselines would be desirable.
- Significance: Given that no faithfulness guarantee can be made about the explanations, it is unclear how to use the interpretations. Without showing that interpretations are sufficiently faithful, the interpretations may not potentially lead to useful applications.

[1] Belrose, Nora, Igor Ostrovsky, Lev McKinney, et al. 2025. “Eliciting Latent Predictions from Transformers with the Tuned Lens.” arXiv:2303.08112. Preprint, arXiv, August 7. https://doi.org/10.48550/arXiv.2303.08112.

---

> ### Author Rebuttal · Authors · 2026-03-31
>
> We thank Reviewer u6Pj for their productive comments. We also appreciate that they highlighted our paper was very well presented and having an intuitive idea. We will now briefly address main concerns and questions:
>
> ## Faithfulness
> First, we would like to emphasize that, next to contributing LatentLens as an interpretability tool, we try to answer and advance a fundamental research question: How can (frozen) LLMs so easily integrate visual input and specifically how do visual tokens relate to the LLM’s inherent representation space?
> This was, and still is, an open question and previous papers either used vanilla EmbeddingLens or LogitLens to make some claims around this question but usually in a less controlled setting and with more limited evaluation, models and comparisons. Overall, we strongly believe this to be a significant scientific question around vision and language representations.
>
> Our paper makes no direct claim about downstream effects but we are optimistic future work will, in a dedicated paper. We did consider focusing on this aspect of downstream behavior and ultimately concluded that the main research question and controlled training study deserve and need the full space of the paper to be most insightful and conclusive to the community.
>
> On a final note, "interpretability" carries different connotations across communities. In mechanistic interpretability, it is often tied to faithfulness. In the VLM literature (e.g. [2,3]), it can also mean whether a representation is interpretable via *another modality* like language. We adopt this definition (p.2): "do their representations correspond to semantically meaningful language?", and operationalize it via our judge design. We will clarify this distinction further in the camera-ready to avoid confusion, and add a disclaimer discussing faithfulness.
>
> ## Baselines
> Reviewer u6Pj asks about stronger baselines:
> To address our main research question we want baselines that are 1) somewhat established for VLMs and 2) come from the inherent embedding space of LLM. Thus, comparing with LogitLens and EmbeddingLens is arguably a fair and reasonable comparison.
>
>  - TunedLens: We are not aware of any VLM paper using TunedLens. Nonetheless, we **investigate further** following Belrose et al. (2023) on four of our model variants, finding: TunedLens does not improve interpretability overall (better on early layers, worse on late). Even if it did, it would only tell us when final-layer predictions are linearly readable, not directly how much visual tokens "look" like language to the LLM at any layer.
>  - SAEs: Non-trivial to compare fairly with our judge setup and require substantial training design choices.
>  - Patchscopes [1]: We explored this initially but found it requires significant prompt engineering and did not work after early layers (confirmed with an author of Patchscopes) We are happy to add it to the paper.
> - Our method is training-free, provides full-sentence interpretations, and allows a flexible corpus (advantages none of these baselines share).
>
> ## Corpus sensitivity
> “What if the text corpus covers only a fraction of possible interpretations, consisting mostly of plausible but not faithful ones?”
> A corpus that covers mostly plausible but not faithful interpretations feels like unusual deployment scenario when users of LatentLens would realistically either adopt our recommended contextual embeddings (we also recently released a more general version of the corpus beyond vision, see response to tiFA) or design their own. But regarding the corpus size, we ran a study to see what happens when the set of contextual embeddings is much smaller (also see response to tiFA) and find that interpretability judgement only starts to drop around 1% of randomly subsampled contextual embeddings.
>
> ## VLM judge bias
> “VLM judge may be biased toward directly observable image concepts”.
> As briefly discussed in the paper (p. 4): “we prompt the judge to determine whether a description is interpretable [...] as concrete (directly visible), abstract (conceptually related), or global [...]”. Manual inspection shows the judge can pick out abstract concepts, e.g.: "The highlighted region shows a [...] face [...]. However, the region can imply emotions or mental states; words like 'introspective', 'thought', and 'thoughtful' relate to the possible mood. [...concludes abstract]". In any interpretability work, it is a general challenge to evaluate complex concepts.
>
> ───
>
> We hope the corpus size ablation and conceptual discussions address your main concerns (happy to discuss further). If so, we would appreciate this being reflected in your rating.
>
> [1] Patchscopes: A Unifying Framework for Inspecting Hidden Representations of Language Models (Ghandeharioun et al., ICML 2024)
>
> [2] Towards Interpreting Visual Information Processing in Vision-Language Models (Neo et al., ICLR 2025)
>
> [3] ClipCap: CLIP Prefix for Image Captioning (Mokaday et al., Arxiv 2021)

---

> > ### Author Rebuttal · Reviewer_u6Pj · 2026-04-02
> >
> > Thank you for your response. It could resolve some of the questions. But the question about faithfulness remains: The authors write
> > > In the VLM literature (e.g. [2,3]), it can also mean whether a representation is interpretable via another modality like language.
> >
> > I wonder what the use of this kind of interpretability is if its faithfulness is not guaranteed. LLMs/VLMs are well known to hallucinate and come up with plausible explanations (including their chain-of-thought). In my opinion, doing causal ablations on one or a few domains would have been necessary for the soundness of the paper. Otherwise, it is difficult to appreciate the results.
> >
> > Having said that, I appreciate the general direction and the breadth of the paper. I can imagine that the paper will become a strong one by addressing the faithfulness issue / by making causal claims, which in my opinion is not an option but a must.

---

> > > ### Author Response · Authors · 2026-04-02
> > >
> > > First, we would like to thank for the reviewer for genuinely engaging with our work and the broader field of interpretability. We as authors have had similar debates over the last year around "What is the point of interpretability/analysis work?" and how can the field improve.
> > >
> > > While it seems the reviewer has mostly made up their mind, which we have to accept, that faithfulness is not just an option, we would like to share briefly our (subjective) perspective and make some case that the paper is a solid scientific contribution:
> > >
> > > When the project was ongoing, several months before submission, we/I were also at this crossroad: Should we focus on downstream performance/outputs of the model, or purely on the main claim/RQ of the paper? We did decide to run downstream experiments primarily testing whether the interpretable visual tokens (using cosine similarity as a proxy) are more important for downstream captioning than the non-interpretable ones. Concretely, we would zero-out e.g. either 30% of interpretable or non-interpretable tokens. However, some LLMs behaved quite strangely and results were too preliminary to share. But more broadly, when the first author reflects on existing analysis-flavored papers, it can seem worse to "squeeze in" some last section in the paper with often very early results but presenting them as "our paper shows that this will improve models/safety". This might have arguably done more harm than good for fields like interpretability (now faced with very high expectations to fix and control models), when authors feel every paper has to claim improvements/safety downstream, instead of simply showing evidence for the identified phenomena. We would rather prefer that some papers (like ours) dedicate themselves to deeply investigating some regularity or pattern in the model, and crucially also ensuring that the results generalize across many models, and many ablations. Then another paper can dedicate not just 1 page but a whole paper to studying downstream effects. As long as the first paper does not claim much about downstream performance, we think this is a fine way for science to advance. For example, hallucination detection is a complicated enough problem that it is hard to focus on in just a small part of the paper, and it should not be done quickly at the end of the project. We think there are many applications of LatentLens that could be more about broad insight than fixing a downstream problem (but hopefully also someone comes and shows if it's useful for e.g. hallucination detection!): Someone studying latent thinking tokens might want to know if these tokens are still close to discrete language throughout the model; or someone studying multi-linguality might want to know at which layers and on which inputs LLMs represent non-English tokens closest to English latent representations (and which ones those are).
> > >
> > >  Finally, in the words of reviewer BNgn: We genuinely think this finding could be a "Big, If True" finding. Thus, we wanted to make sure that it is in fact true and laser-focus on that alone: with controlled training setup, ablations, manual inspection of many examples, demo, etc. Even then, reviewers asked for more models and other empirical justifications, showing us that we can do better even in this regard (e.g. now having 3 off-the-shelf models and not just 1, see BNgn response). Often, interpretability papers can fall short here instead of faithfulness: using only 1-2 off-the-shelf models, not ablating the training (understandably with the costs).
> > >
> > > We understand that the reviewer has this strong and justified focus on faithfulness (the field has had many issues with it in the past), and there might just be a fundamental divide in opinions here. We hope sharing our perspective was at least interesting and maybe helpful.

---

### Official Review · Reviewer_khVQ · 2026-03-12

**Soundness:** 3
**Presentation:** 2
**Significance:** 3
**Originality:** 3
**Overall Recommendation:** 5
**Confidence:** 3

**Summary:**

The authors introduce LatentLens, a new way to interpret visual tokens. LatentLens works by building a corpus of LLM embeddings from different layers of a VLM-trained model that correspond to text sequences from a corpus. Then, vision tokens are interpreted by computing cosine-similarity to this corpus of text sequence embeddings across different layers & gathering the top-5 closest matches.

The main contributions are a new strategy for interpreting vision tokens (resulting in more interpretable text alignment than LogitLens and EmbeddingLens) and interesting analysis about the properties of vision tokens – particularly that they may be natively postprocessed by the projection head to align with the middle layer(s) of an LLM.

**Compliance With Llm Reviewing Policy:**

Affirmed.

**Final Justification:**

Authors adequately addressed all concerns, resolving concerns about evaluation and scale. I raise my rating to a 5.

**Key Questions For Authors:**

How long it takes to construct the embedding corpus, how big is the storage footprint, how long does it take to identify the NNs?

Is there a separate corpus per trained model? How does interpretability change over training for the three methods? Are they similarly interpretable at the start, what are the slopes as training progresses, etc.

Why do you think DINOv2 is the worst for Qwen and Llama but works well for OLMo for EmbeddingLens and LogitLens?

**Limitations:**

yes

**Strengths And Weaknesses:**

Soundness: Experiments are convincing, especially Figure 3 showing interpretability across layers. The fact it works in off the shelf models is also impressive. My main concern though is the method of evaluating interpretability, effectively taking pass @ 5 over the nearest neighbors, and the use of GPT-4o for evaluation. Results with narrower precisions and a stronger VLM would be nice to verify this improvement.

Presentation: Work is very well written and presented. Relevant literature is discussed (although I am not the most familiar).

Significance/Originality: I think this work does address a relevant problem - modern VLMs that tie vision encoders to LLMs is counterintuitive and not well understood and examining vision tokens in LLMs gives insight into not only why this works, but also the nature of LLMs themselves. This work does a good job highlighting that vision tokens are comparable to mid-layer embeddings of longer sequences, encoding more complex and abstract information than language tokens.
One weakness is the use of a corpus. It would be good to see ablations on corpuses of different sizes (mainly larger).

---

> ### Author Rebuttal · Authors · 2026-03-31
>
> We thank reviewer khVQ for their review that shows they engaged with the paper in-depth. We are happy they align with us on being excited by the “mid-layer leap” phenomenon we identified. We appreciate that the reviewer considers our experiments “very convincing” and the paper “very well presented”, while addressing a “relevant problem”. We will now briefly address their concerns as well as further clarification questions:
>
> ## VLM judge evaluation
> "[...] Results with narrower precision (not pass@5) and a stronger VLM would be nice”
> A small clarification: our final experiments in the paper use GPT-5 as the judge, not GPT-4o. We apologize for the confusion in the original text and have fixed this.
> Regarding the pass@5 concern, we investigated pass@1 instead of pass@5 and observe avg. interpretability drop from 72% to 49% on LatentLens. We manually inspect 10 examples where the judge previously said “interpretable” and now not anymore:
>
> Example of top-5 LatentLens result (image of a man in a gym working out, with a lamp not too far from him)
> 1. off, — "street light appears to be *off* [...]"
> 2. workout — "guy wearing *workout* clothes"
> 3. workout — "[...] man stretching before a *workout*"
> 4. fitness — "[...] good presentation for a *fitness* model."
> 5. gymnastics — "young athletic man doing *gymnastics* by [...]"
>
> **Conclusion:**
> Due to fair comparison to the other 2 methods (and difficulties calibrating precision/recall), we only show the judge the full word but not the sentence it appeared in. Thus, showing only top-1 increases the probability that one word alone is not enough to see the semantic connection, such as "off" above. Overall, pass@1 does decrease results but from inspection the majority of this is driven by judge design of only seeing a single word and not an actual large drop in interpretable tokens.
> Also, LatentLens at pass@1 (49%) still beats both baselines at pass@5 (30% and 19%).
> We acknowledge designing this judge has its difficulties but note that previous studies rely on more restrictive methods altogether such as matching a fixed list of class words [1].
>
> ## Corpus ablation
> Regarding the corpus size, we run an ablation on 3 models (OLMo+CLIP, LLaMA+SigLIP, Qwen2+DINOv2) and find that interpretability judgement only starts to drop significantly around random subsampling of 1% of the embeddings (from 72% to 67%). See response to R1 tiFA (who had a similar question) for a detailed table.
> We also have a proof-of-concept for a corpus generated on-the-fly via LLM-guided evolutionary search (also see response to tiFA).
>
> ## Implementation details
> For a detailed breakdown, see our response to tiFA. The summary is: between 2-13h to compute+store the contextual embeddings once; 26GB to store all contextual embeddings for a given model (but we have seen that the corpus can be much smaller); fast inference at around 30ms per image. We will add this to the paper.
>
> "Is there a separate corpus per trained model?" The corpus is the same but the computed contextual embeddings change per LLM backbone. In our case we had three frozen LLM backbones.
>
> ## Training dynamics
> “How does interpretability change over training for the three methods? [...]”
> This is a great question! Since our automatic metric is somewhat costly, we did not study this in-depth but had considered it. For this rebuttal, we re-ran training of the MLP connector for our OLMo+CLIP variant and found something interesting:
>
> | Step | Avg % interpretable (LatentLens) |
> |------|--------------------------------|
> | 100 (0.8K examples) | 11.7% |
> | 1000 (8K examples) | 28.1% |
> | 6000 (48K examples) | 75.1% |
> | 12000 (96K examples) | 72.3% |
>
> At step 1000, only the earliest layers (LLM layer 0-1) become interpretable (~66-68%) while deeper layers remain at 14-20%. By step 6000, all layers converge to near-final performance. We will add this finding in the camera-ready and expand it to LogitLens and EmbeddingLens.
>
> ## DINOv2 performance across LLMs
> “Why do you think DINOv2 is the worst for Qwen and LLaMA but works well for OLMo for EmbeddingLens and LogitLens?”
> This is also a good question. Beyond DINOv2, it is also not clear why EmbeddingLens works so much better in general on Olmo as the LLM backbone. It does not correlate with captioning performance (Appendix I, Caption Quality Evaluation).
> We should note that LogitLens is known to work inconsistently across model families [2], and we are not aware of a good explanation for why the (un)embedding matrix works better for some models than others.
>
> ───
>
> We hope the corpus size ablation, the pass@1 experiment and further conceptual discussions address your main concerns. If so, we would appreciate this being reflected in your rating.
>
> [1] Towards Interpreting Visual Information Processing in Vision-Language Models (Neo et al., ICLR 2025)
>
> [2] Eliciting Latent Predictions from Transformers with the Tuned Lens (Belrose et al., Arxiv 2023)

---

> > ### Author Rebuttal · Reviewer_khVQ · 2026-04-02
> >
> > I thank the authors for their response. Most of my initial concerns are resolved. I am a little wary of the time it takes to construct the corpus, and whether this would still work at scale for larger models >30B where some step function might apply to model capabilities, especially for jointly trained Encoder and LLMs as pointed out by reviewer BNgn.
> >
> > I maintain my score for now but am curious to hear responses to these points.

---

> > > ### Author Response · Authors · 2026-04-06
> > >
> > > Thank you for further engaging with our work.
> > >
> > > **Time to compute+store contextual embeddings:**
> > >
> > > We like to frame our approach primarily as a paradigm: **Contextual embeddings are the better space to compare in** (and not input or output level embedding matrices). This paradigm can be instantiated in many ways and our pool of embeddings derived from 3M VG phrases was a proof-of-concept suitable to our RQ.
> > > In this rebuttal we have already shown that even with just 1% of the contextual embeddings (randomly sampled), it works almost as well. Results pasted here again:
> > >
> > > | Percentage of subsampled embeddings | Avg % of interpretable visual tokens |
> > > |-------------|-------------------|
> > > | 0.1% | 58.4% |
> > > | 1% | 67.3% |
> > > | 10% | 72.5% |
> > > | 100% | 71.6% |
> > >
> > > We see LatentLens with a massive corpus simply as an upper ceiling: "In the ideal scenario, this is how interpretable visual tokens can be. If you want to get very quick results, you can adopt a faster and smaller implementation and get very good results still."
> > >
> > > So to actually help get people started in practice: We have already released a much more curated and smaller corpus with 119K sentences, derived from 23K WordNet concepts (roughly, an LLM generated sentences for each these concepts). This will run much faster and we released a pip library where embedding creation takes a few lines.
> > >
> > > **Larger models:**
> > >
> > > We agree: Interpretability insights should be shown to generalize across many settings: We have shown controlled training variants and also off-the-self models, as well as several ablations across unfreezing, mapping network or training data. Since model size is another good axis, we followed your suggestion and also investigated threee models above 30B parameters: **Qwen2.5-VL-32B, LLaVA-NeXT-34, Molmo-72B**. We find again similar trends of LatentLens clearly outperforming the other two baselines overall:
> > >
> > > | Model | EmbeddingLens | LogitLens | LatentLens (Ours) |
> > > |-------------|---------------|-----------|-------------------|
> > > | Molmo-72B | 70.3% | 29.8% | **78.4%** |
> > > | Qwen2.5-VL-32B | 12.5% | 11.3% | **35.1%** |
> > > | LLaVA-NeXT-34B | 28.3% | 26.5% | **33.2%** |
> > >
> > > However now we notice more fluctuating behaviors for two of the models: For Qwen2.5-VL-32B the mid and later layer interpretability drops, while for LLaVA-NeXT-34B we observe a stark drop at layers 1 to 8 (but not layer 0 and later layers).
> > > We will discuss large models (and also might try to identify an initial explanation) in our updated paper. Importantly, we do think that the overall message of the paper still holds: in most cases the intermediate representation space is more meaningful to compare against, and not input or output embeddings. LatentLens work much more consistently across almost all settings/layers than LogitLens or EmbeddingLens; We note that LogitLens, as a similar precedent, is widely used and popular despite being known to not work on some LLMs in general (see: "Eliciting Latent Predictions from Transformers with the Tuned Lens", Belrose 2023).

---

### Official Review · Reviewer_tiFA · 2026-03-12

**Soundness:** 3
**Presentation:** 2
**Significance:** 2
**Originality:** 3
**Overall Recommendation:** 4
**Confidence:** 3

**Summary:**

This paper studies whether visual tokens inside vision language models are understandable in language space. The authors propose a method called LatentLens. The idea is simple. They collect many contextual token representations from a large text corpus. Then they compare visual token representations with these contextual text representations using nearest neighbors. The retrieved neighbors are used as natural language descriptions of the visual tokens. The authors test this method on ten different vision language models. They find that many visual tokens are actually interpretable when using this approach. Previous methods such as LogitLens or EmbeddingLens often show very poor interpretability. LatentLens instead reveals meaningful descriptions for most visual tokens across layers. The results suggest that visual and language representations are more aligned than previously believed.

**Compliance With Llm Reviewing Policy:**

Affirmed.

**Final Justification:**

Raised my score from 3 to 4 due to the author's detailed and responsible rebuttal.

**Key Questions For Authors:**

How sensitive is LATENTLENS to the coverage of the text corpus used to build the contextual embedding database? What happens if the corpus is smaller or from a different domain?


Have the authors tried using the interpretations from LATENTLENS to improve models or assist downstream tasks, such as debugging VLM errors or improving visual grounding?


What is the tradeoff between corpus size and interpretability performance? Is there a point where adding more contextual embeddings gives little improvement?


How expensive is the nearest neighbor search in practice when the contextual database becomes very large?

**Limitations:**

The method relies on a large external text corpus. If the corpus does not contain relevant contexts, the interpretation may be inaccurate.


The approach mainly provides analysis and interpretation. It does not directly improve model performance or solve downstream tasks.


The method requires storing and searching a large database of contextual embeddings, which may limit scalability.


The evaluation of interpretability mainly relies on a VLM judge, which may introduce bias in the interpretation quality.

**Strengths And Weaknesses:**

**Strengths**:

The motivation is interesting. It tries to solve a concrete problem. Previous work suggests visual tokens are not very interpretable. This paper questions that assumption and studies it more carefully.


The experiments are quite thorough. The authors test the method on many VLMs and compare with several baselines. They also show some useful analysis across layers.


The results are fairly strong. The method shows that many visual tokens can actually be interpreted. This gives some new insight about the alignment between vision and language representations.

**Weaknesses**:

The method depends on a large contextual text corpus. If the corpus does not cover enough contexts, the nearest neighbors may not describe the visual token well. The paper does not study how sensitive the method is to the corpus coverage.


It is not clear how useful this method is for downstream tasks. The paper mainly focuses on analysis. It would be helpful to show whether this interpretation signal can help model improvement, debugging, or other practical tasks.


The approach requires storing a very large number of contextual embeddings. This may be expensive in memory and computation. The paper does not analyze how the corpus size affects performance or what the minimal size is.

---

> ### Author Rebuttal · Authors · 2026-03-31
>
> We thank Reviewer tiFA for their comments and questions. We also appreciate that they highlighted the “interesting motivation” as well as our strong and in-depth empirical results. We address three main concerns or questions below:
>
> ## Corpus sensitivity
> First, we think that it is actually a strength of our method that you can customize the corpus. Beyond vision, someone else might customize it for multi-linguality or reasoning.
> But we do think the paper and future research would benefit from an ablation, where we consider only a subset of the full contextual embeddings. Thus, we conducted an additional experiment for camera-ready: Instead of using 100% of the contextual embeddings derived from Visual Genome sentences, we use either 10%, 1% or 0.1% of the contextual embeddings.
>
> We run this ablation on 3 models (OLMo+CLIP, LLaMA+SigLIP, Qwen2+DINOv2) and find that interpretability only starts to drop significantly below 1% of the embeddings (averaged across 3 models and 4 LLM layers each):
>
> | Corpus size | Avg % of interpretable visual tokens |
> |-------------|-------------------|
> | 0.1% | 58.4% |
> | 1% | 67.3% |
> | 10% | 72.5% |
> | 100% | 71.6% |
>
> (small increase from 100% to 10% due to sampling noise with judge on 100 image patches)
>
> These results relate to an intuition from the paper about fine-grained interpretability: A smaller corpus would not really reduce whether a visual token is deemed interpretable, but it will make the interpretation less fine-grained or precise on a sentence level.
> So to answer the reviewer’s question directly: The saturation point seems to be around 1% for the *binary* question whether a token is deemed interpretable. But quality of interpretation can benefit from larger corpora.
>
> ## Regarding the reviewer’s question about different domains for the corpus
> We thank the reviewer for this question, as it in fact highlights one of the core strengths of LatentLens: the corpus is fully user-controlled. Unlike approaches tied to fixed embedding spaces (e.g., using an LLM's unembedding matrix), LatentLens allows you to bring domain expertise directly into the interpretability pipeline by choosing a corpus relevant to your setting!
>
> That said, a domain-specific corpus is not required. Any sufficiently large and diverse corpus will yield useful results. To demonstrate this, we constructed and evaluated a general-purpose corpus of 117k sentences spanning ~23k concepts, derived by intersecting WordNet lemmas with Brown Corpus vocabulary — providing broad, common-usage coverage beyond visual concepts. We will release pre-built indices for several models alongside the camera-ready version, so users can get started without constructing their own corpus at all.
>
> We will also describe a dynamic/automatic corpus approach (proof-of-concept already available) where sentences are generated on-the-fly via LLM-guided evolutionary search, iteratively refining candidates based on cosine similarity.
> In summary: domain-specific corpora are a feature, not a requirement. We will make this flexibility explicit in the revised paper.
>
> ## Downstream effects
> We were deliberate about making the fundamental research question and deep analysis the central part of the paper. We think the question of how frozen LLMs can integrate visual tokens into its own linguistic space is fundamental enough to deserve a deep dive. Every paper before us usually uses only few off-the-shelf models and does not compare these different ways of finding nearest neighbors.
> We are optimistic that there will be interesting downstream effects or use cases, similar to how Logitlens has helped for hallucination detection [1], serving as a diagnostic tool for hallucination or grounding failures. And we are equally optimistic that it will also inform other interpretability efforts where LLMs process inputs like action tokens in Vision-Language-Action models, audio tokens or latent thinking tokens.
>
> ## Practical effort to run LatentLens
> We thank the reviewer for raising this point and will highlight more of the implementation details in the paper:
>
> 1. Corpus construction (one-time per LLM): The forward pass through \~3M sentences takes \~2 hours of GPU compute (on an RTX A6000, batch size 64). Total wall time including index building is \~13h (this could be done more efficiently than our vanilla approach).
> 2. Storage of contextual embeddings: \~26 GB per model (summed across 8 LLM layers of OLMo-7B, float8). Our corpus ablation shows 1% of embeddings gives comparable interpretability scores, with just \~250 MB. Using a more concise corpus can also help.
> 3. Inference speed: Once the index is loaded into memory, NN search takes \~29ms per image (on RTX A6000).
>
> ─
>
> We hope the corpus ablation results and efficiency numbers directly address your main concerns (W1, W3). If so, we would appreciate this being reflected in your rating.
>
> [1]: Interpreting and Editing Vision-Language Representations to Mitigate Hallucinations, Jiang et al., ICLR 2025.

---

> > ### Author Rebuttal · Reviewer_tiFA · 2026-04-03
> >
> > Thanks for the detailed response, I will raise my score from 3 to 4 in response.

---

### Decision · Program_Chairs · 2026-04-30

**Decision:**

Accept (regular)

**Comment:**

This paper introduces LatentLens, a new method for interpreting the visual tokens in Vision Language Models. It had fruitful discussions during the rebuttal. And the paper received scores of 3445 by the end. On the positive side, reviewers have commented that the motivation is interesting, the experiments are quite thorough, and the results are fairly strong. On the other hand, reviewers also raised concerns regarding faithfulness of the generated explanations, stronger baselines, corpus sensitivity and VLM judge bias. Overall, the AC thinks that the authors have done a good job of rebuttal, and the merits outweigh the flaws, and would like to recommend acceptance by the end.